# Cellular location shapes quaternary structure of enzymes

György Abrusán [1] ✉ & Aleksej Zelezniak[1,2,3]

The main forces driving protein complex evolution are currently not well understood, especially in homomers, where quaternary structure might frequently evolve neutrally. Here we examine the factors determining oligomerisation by analysing the evolution of enzymes in circumstances where homomers rarely evolve. We show that 1) In extracellular environments, most enzymes with known structure are monomers, while in the cytoplasm homomers, indicating that the evolution of oligomers is cellular environment dependent; 2) The evolution of quaternary structure within protein orthogroups is more consistent with the predictions of constructive neutral evolution than an adaptive process: quaternary structure is gained easier than it is lost, and most extracellular monomers evolved from proteins that were monomers also in their ancestral state, without the loss of interfaces. Our results indicate that oligomerisation is context-dependent, and even when adaptive, in many cases it is probably not driven by the intrinsic properties of enzymes, like their biochemical function, but rather the properties of the environment where the enzyme is active. These factors might be macromolecular crowding and excluded volume effects facilitating the evolution of interfaces, and the maintenance of cellular homeostasis through shaping cytoplasm fluidity, protein degradation, or diffusion rates.

The majority of proteins form complexes, however, the main forces driving quaternary structure evolution are currently not well understood, especially in the case of homomers - complexes made from multiple units of the same protein. Traditionally, protein multimerization was viewed as the result of adaptation[1,2], however, in recent years, it was demonstrated that neutral processes also significantly influence the evolution of quaternary structure[3–6], and currently it is unclear what is the relative importance of adaptive vs. stochastic factors in protein complex evolution.

In many complexes, multimerization contributes to their biochemical function, which can be related to allostery[7–10], gene regulation[11,12], structural stabilisation[13] or other factors[14]. In their pioneering (and quite overlooked) work, Chan and colleagues have demonstrated that in some enzymes which do form stable homomers,

the subunits can nevertheless perform their catalytic functions in isolation almost as well as in complex[15–17], and hypothesised that in such cases, the primary role of multimerization is not related to the biochemical function, but to other factors like proteostasis, compartmentalisation, or osmotic regulation. The importance of multimerization in proteostasis was demonstrated in recent years[18], while we only begin to understand the role of protein condensates in maintaining cellular water availability[19], even though the role of proteins in maintaining osmotic pressure is well studied[20].

However, in several homomers, the contribution of quaternary structure to any function is questionable[21–23], and it was suggested that neutral processes can also significantly influence their evolution. Lynch has pointed out that, unlike many genomic traits, protein oligomerisation is not correlated with the strength of selection, and in

[1]Randall Centre for Cell and Molecular Biophysics, School of Basic and Medical Biosciences, King's College London, New Hunt's House, London, UK.
[2]Department of Life Sciences, Chalmers University of Technology, Gothenburg, Sweden. [3]Institute of Biotechnology, Life Sciences Centre, Vilnius University, Vilnius, Lithuania. ✉e-mail: gyorgy.abrusan@kcl.ac.uk

**A)** Mus musculus   **B)** E. coli   **C)** A. pleuropneumoniae

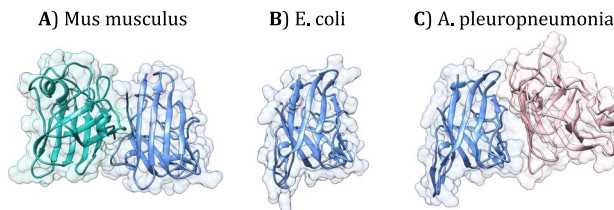

**Fig. 1 | Examples of quaternary structure variability in homologous Cu-Zn Superoxide dismutases (CuSOD, EC = 1.15.1.1). A** In eukaryotes, e.g., mouse (PDB ID: 3GTT), CuSOD is a homodimer with a small but highly conserved interface. **B** In prokaryotes, it is either a monomer (*E. coli*, PDB ID: 1ESO) or (**C**) a homomer (e.g., *Actinobacillus pleuropneumoniae*, PDB ID: 2APS). However, the interfaces in prokaryotes are not homologous with the interfaces of eukaryotes. (Note that on the three panels, the blue subunits are structurally aligned).

many homologous enzymes with the same biochemical function, the quaternary structure is different (see Fig. 1 for an example), and suggested that neutral forces are likely to play a significant role in the evolution of quaternary structure[3,24]. This was also supported by other theoretical findings showing that protein interactions can evolve without providing a fitness advantage[25]. Subsequently, using the change in ligand binding ability as a measure of functional divergence, we performed a PDB-wide analysis to test whether quaternary structure influences the evolution of new functions[4,26]. We found that complexes where the binding sites are restricted to a single protein chain, are not different from monomers, and also their folds have similar characteristics[26], which suggests that quaternary structure frequently does not influence the evolution of biochemical function and that in these complexes neutral evolution is frequent.

Recently, Hochberg et al.[5] provided experimental evidence that in steroid receptors, dimerisation does not have a measurable contribution to function, and proposed that the evolution of protein interfaces follows a ratchet-like pattern. These authors suggested that the driving mechanism behind the long-term "entrenchment" of interfaces is constructive neutral evolution (CNE)[27,28]: the high frequency of hydrophobic residues in random mutations can create interfaces randomly, which can be deleterious when exposed to solvent (due to aggregation or destabilisation), and thus can be maintained by purifying selection without providing any fitness advantage[5]. A subsequent bioinformatic analysis by us has demonstrated that the hydrophobicity of interfaces is largely independent of the strength of selection acting on them, indicating that it is maintained by a neutral mechanism[6] and that in ~35% of homodimers with small to medium interface size, the interface may not contribute to the biochemical function of the protein[6].

In this work, we test the predictions of the CNE, and suggest that macromolecular crowding and the maintenance of cellular homoeostasis are also key factors in the evolution of homomers. Using monomer and homomer enzymes from the Protein Data Bank (PDB), we examine the factors determining oligomerisation by analysing the evolution of proteins in situations when homo-oligomers rarely evolve, or their interfaces are lost. We hypothesised that in different cellular environments, the probability of the evolution of homomers is different: in the crowded cytoplasm, the probability of protein associations is high due to the high frequency of interactions and the excluded volume effect[29,30], that, in the long-term, might favour the evolution of new interfaces. In contrast, in the extracellular environment, the frequency of complexes is likely to be low due to several mechanisms: the low concentration of subunits preventing assembly or resulting in their dissociation, difficulties in transporting assembled complexes in Prokaryotes, or the lower diffusion rates of large oligomers (Fig. 2A).

Our findings confirmed this hypothesis, and we show that extracellular enzymes are predominantly monomers, while in the cytoplasm, homomers are prevalent. In addition, we found that the

abundance of homomers is consistently higher in the cell than the abundance of monomers, indicating that high protein abundance facilitates homomer evolution. The analysis of quaternary structure evolution within orthologous proteins shows that quaternary structure is gained easier than lost and that the evolution of extracellular proteins is largely consistent with the predictions of CNE, i.e., a unidirectional, ratchet-like evolutionary process, most likely due to the hydrophobic ratchet, but we hypothesise that a degradation-based ratchet also contributes to the pattern. Taken together, our findings indicate that the factors influencing oligomerisation are significantly shaped by the properties of the cellular environment, like macromolecular crowding, and not only by the intrinsic properties of proteins (i.e., their biochemical function), and that oligomerisation is likely to contribute to the maintenance of cellular homoeostasis.

## Results
### Data sources
We have extracted all proteins from the PDB, which are either monomers or homomers; their structure contains at least 80% of their UniProt sequence and has a resolution better than 3 Å (see "Methods" for details). Proteins that are also part of heteromer PDB structures were not included. Next, the sequences were filtered for enzymatic activity, and only the ones with a known Enzyme Commission (EC) number or for which an EC number could be reliably predicted were kept in the dataset (see "Methods"). This resulted in a list of 11968 enzymes (Supplementary Data 1), of which 8248 are homomers, and 3720 are monomers. Analyses of surface hydrophobicity, absence of transmembrane helices, and known heteromer complexes indicate that the quality of the dataset is high (Supplementary notes and Supplementary Figs. 1 and 2). Cellular location was determined with DeepLoc2[31] for Eukaryotes and PSORTb[32] for Prokaryotes, resulting in 7562 cytoplasmic, 894 extracellular, and 1726 other proteins. As the cellular compartments and extracellular environments of Prokaryotes and Eukaryotes are frequently very different, just as their protein secretion/transport mechanisms, wherever it was possible, we analysed Prokaryote and Eukaryote proteins separately (8228 and 3740 proteins, respectively). To remove redundancies in Figs. 2 and 3, we clustered the proteins with MMseqs2[33] at a 30% sequence similarity cutoff. This resulted in 2435 prokaryotic and 1528 eukaryotic clusters for which the cellular location could be determined, and the cluster centroids were used in the downstream analyses.

### Extracellular enzymes are predominantly monomers
The analysis of prokaryotic and eukaryotic proteins indicates that despite the huge differences in their biology and composition of their extracellular/intracellular space, enzymes in the two groups show a largely similar pattern: in the cytoplasm, homomers dominate (70% of the clusters, Fig. 2B), while in the extracellular space monomers (70% of the clusters, Fig. 2B), and proteins of other cellular components have intermediate frequencies of homomers and monomers (Fig. 2B). The differences in the frequencies also result in qualitative differences in homomer interfaces: in homodimers, the fraction of the protein surface that is buried in interfaces is much smaller in the case of extracellular proteins than in cytoplasmic ones (Fig. 2C), and in extracellular proteins, the interface is also much less conserved compared to the protein surface (Fig. 2D–F). Conservation was measured as the difference between the ConSurf scores[34] of the interface and surface residues. Homomers with small interfaces, particularly below 1000 Å², are likely to be enriched in quaternary structure (QS) errors[6,35] (i.e., can have non-biological, crystallographic interfaces), thus the real frequency of monomers in extracellular proteins is probably even higher, and our estimates are conservative. When only homomers with 1000+ Å² interfaces are included, in the cytoplasm, 30.2% of the clusters have interfaces that are not significantly more conserved than the

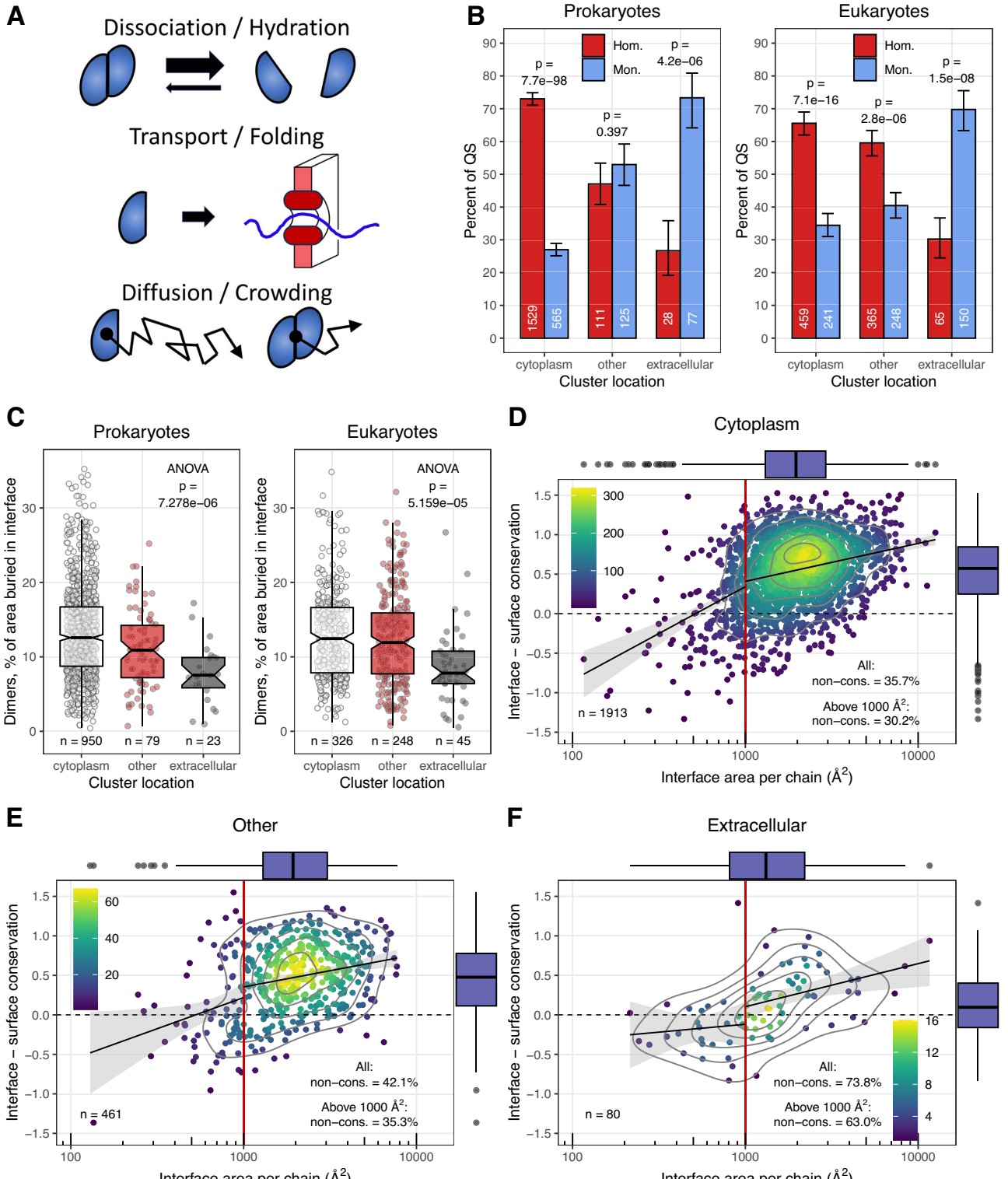

**Fig. 2 | Extracellular enzymes are predominantly monomers. A** Mechanisms that might prevent the evolution of oligomers in extracellular enzymes: low concentration of subunits resulting in dissociation, limitations of transport mechanisms (in Prokaryotes), and selection for high diffusion rates. **B** Extracellular enzyme clusters are dominated by monomers, while cytoplasmic ones by homomers, both in prokaryotes and eukaryotes. Bars indicate the fraction of quaternary structure in the set, whiskers represent 95% CI. *P*-values were calculated with tests of proportions and were corrected with the Benjamini-Hochberg method. **C** The relative size of homomer interfaces is largest in the cytoplasm, and smallest in the extracellular space. **D–F** The conservation of homomer interfaces (relative to the solvent-accessible surface) shows a similar trend as their size and frequency: it is highest in the cytoplasm, and lowest in the extracellular space, where the fraction of non-conserved - and probably non-functional - interfaces is very high, 63.0–73.8%. As non-biological (crystallographic) interfaces in the PDB are primarily present in homomers with interface areas below 1000 Å²; and prokaryotes and eukaryotes were pooled. On panels (**C–F**) boxplots display the median, 25–75% interquartile range (IQR), and 1.5 * interquartile range from the hinge (whiskers). Notches, when present, are defined as 1.58 * IQR / sqrt(n). Datapoints beyond the whiskers are shown as outliers.

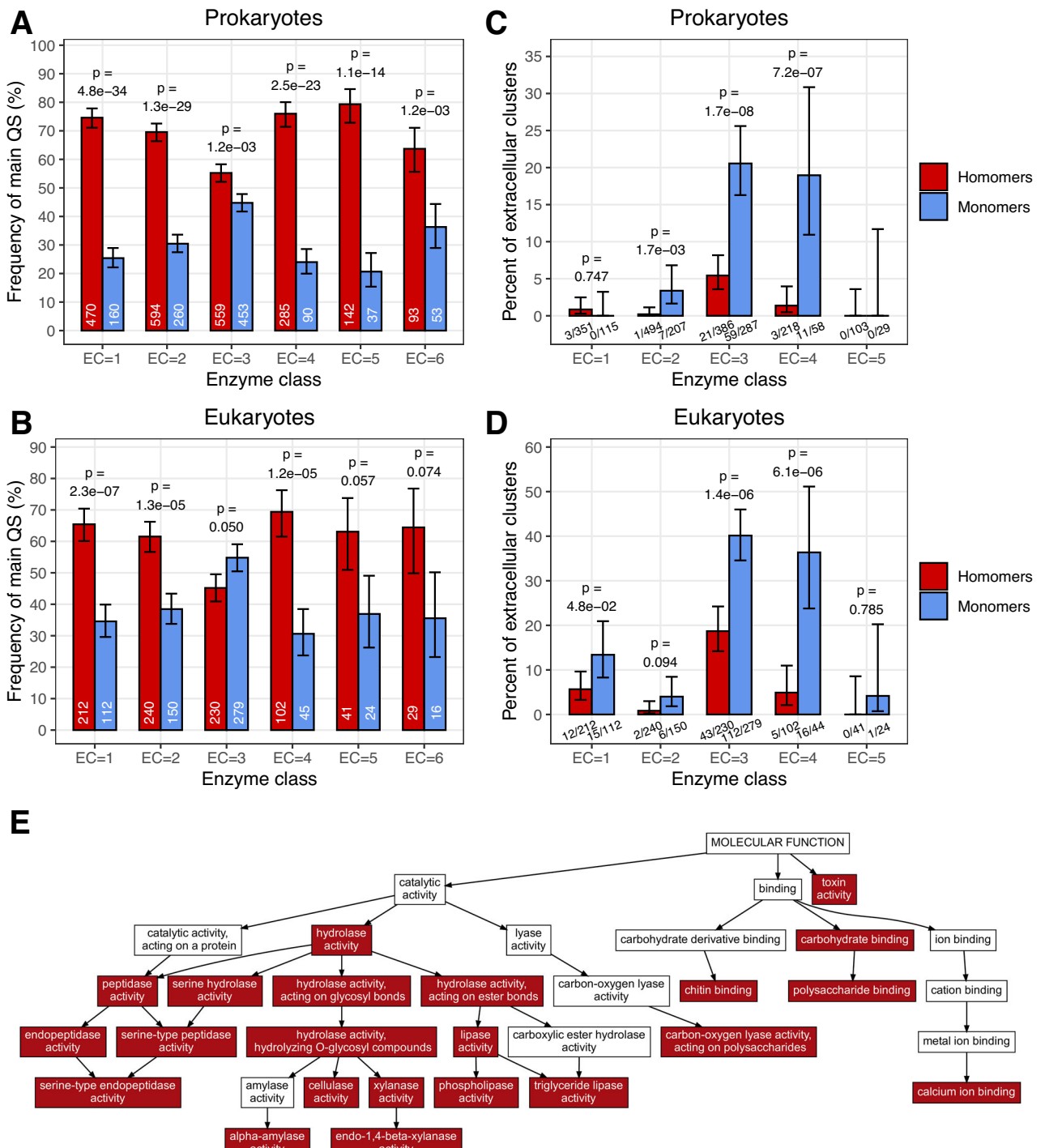

**Fig. 3 | Extracellular enzymes are the most frequent among hydrolases.**
**A**, **B** Monomers are most common in hydrolases (EC = 3), while in other enzyme classes, homomers dominate, and their homomer/monomer ratio is between 2–3 in prokaryotes and 1.5–2 in eukaryotes. **C**, **D** The frequency of extracellular enzymes is comparably high among the monomers of hydrolases (EC = 3) and lyases (EC = 4), both in prokaryotes and eukaryotes. In absolute terms, however, extracellular hydrolases are much more abundant in the PDB: in total, there are 1521 hydrolase clusters in the dataset, while only 530 lyase clusters and the frequency of monomers is also much lower in lyases. **E** Graph of Molecular Function GO terms, the terms that are significantly enriched (hypergeometric test) both in prokaryotes and eukaryotes are highlighted with red. The Bonferroni method was used to correct for multiple testing. A diverse array of hydrolase activities, but also carbohydrate binding, toxin activity, or lyase activity on polysaccharides are enriched in both groups. (See Supplementary Data 1 for the full lists of Molecular Function, and Biological Process GO terms.) On panels (**A**–**D**), p-values were calculated with tests of proportions and were corrected with the Benjamini-Hochberg method; bars indicate the fraction of quaternary structure in the set, and whiskers represent 95% CI.

solvent-accessible surface (Fig. 2D) while in the extracellular space, 63.0% (Fig. 2F), which is in good agreement with our recent findings showing that in ~35% of homodimers, multimerization is unlikely to contribute to the biochemical function[6].

Taken together, these findings are in agreement with the hypothesis that in the extracellular space, oligomerisation is either difficult or deleterious (Fig. 2A), and that results in the evolution and dominance of monomer enzymes. Since the pattern is essentially

similar in eukaryotes and prokaryotes, it is unlikely to be caused by the limitations of the protein secretion mechanisms of prokaryotes, as eukaryotes can secrete large, assembled complexes. We add that in the final stages of revisions we found out that Monod has already noticed that extracellular proteins are usually monomers[36] and hypothesised that it is due to their stabilisation by disulfide bridges. However, this explanation can be ruled out due to the much higher frequencies of disulfide bridges in eukaryotes than in prokaryotes, and the fact that quaternary structure varies in enzymes with the same folds and function.

## Extracellular enzymes are most common in hydrolases and lyases

Next, we examined whether monomers and extracellular enzymes are enriched in particular enzyme classes. We found that monomers are most abundant among hydrolases (EC = 3), where their frequency is comparable to the frequency of homomers (Fig. 3A, B), while in the case of other enzyme classes, the ratio of homomers to monomers is much higher, ~2–3 in prokaryotes and ~1.5–2 in eukaryotes (Fig. 3A, B). The frequency of extracellular enzymes is highest in the monomers of hydrolases and lyases (EC = 3 and 4, Fig. 3C, D), and the difference between homomers and monomers remains highly significant within these enzyme classes, both in Eukaryotes and Prokaryotes, suggesting that the majority of extracellular enzymes in the PDB are involved in the degradation of macromolecules (Fig. 3C, D).

In addition to their distribution in enzyme classes, we also examined the enrichment of Gene Ontology terms in extracellular proteins, compared to the full dataset (Fig. 3E and Supplementary Data 2). Similarly to EC numbers, hydrolase activity is highly enriched among molecular function terms, both in Eukaryotes ($p$ = 2.5e-144) and Prokaryotes ($p$ = 2.9e-47, Fig. 3E and Supplementary Data 2), and all major hydrolase forms are enriched in the dataset (hydrolyses acting on glycosyl bonds, peptidases, lipases, serine hydrolases; Fig. 3E and Supplementary Data 2). The "biological process" terms are less comparable in eukaryotes and prokaryotes due to their different complexity, but in both groups "catabolic process" and its derived terms are the most frequently enriched (see Supplementary Data 2).

The above results indicate that hydrolases are the most common enzymes in the extracellular space, however, hydrolases also represent 75% of industrially relevant enzymes[37], which may result in research biases in the composition of PDB. Thus, we examined whether excluding them from the data changes the pattern, i.e., whether the enrichment of monomers is a hydrolase-specific phenomenon or a more general trend. We found that their exclusion does not change qualitatively the patterns we observe (Supplementary Fig. 3), although significances are affected, due to the much lower number of clusters (e.g., in eukaryotes, the total number of extracellular clusters drops from 214 to 58, while in prokaryotes to 25, from 105).

## The evolution of interfaces follows a ratchet-like pattern

Next, we examined whether the evolution of homomers from monomers (or vice versa) shows any biases, i.e., whether interfaces are more likely to be gained than lost. Using the full set of 12 k eukaryotic and prokaryotic sequences, we identified orthogroups within them using the eggNOG-mapper tool[38] (see "Methods"). Altogether, 311 orthogroups had ten or more sequences, and these were used in the downstream analyses. The proteins were aligned, and their rooted phylogenetic trees were used to estimate the evolution of oligomeric status along the phylogeny (see "Methods"), i.e., the ancestral probabilities of being a monomer or homomer for each node (Fig. 4A, see pie charts at each node). Most orthogroups have ancient proteins: 243 have proteins from more than one domain and 92 from all three domains (Bacteria, Archaea, Eukaryota). Ancestral probabilities were estimated in two ways: a binary, and the more accurate multi-state method. In the binary case, proteins were

assigned to one of two states: monomers or homomers, irrespective of the number of their subunits. In the multi-state case, the number of subunits (monomer, dimer, tetramer, octamer, etc.) was used as the discrete variable for ancestral character estimation, and we also took into account the homology of interfaces: complexes with similar number of subunits but non-homologous interfaces were treated as separate states in the trees. The estimation of ancestral state was performed with maximum likelihood (ML, Figs. 4 and 5) and also with maximum parsimony (MP, Supplementary Figs. 4 and 5), which show similar results. Trees with only a single quaternary structure type (e.g., all proteins being homomers) were included, but trees where the highest ancestral probability at the root was lower than 51% were excluded from the analyses (31 binary and 57 multi-state trees with ML, and 37 binary and 78 multi-state trees with MP).

To test whether there are trends in the evolution of homomers, we split the trees into two groups depending on whether the root was a homomer or a monomer (Fig. 4A). Next, we examined the frequency of quaternary structure changes in the trees. We found large differences between the phylogenies: in the binary analysis, in the trees where the ancestral state was a monomer, the frequency of multimerization and the fraction of homomers in the tree is much higher than the frequency reversion to a monomer in those trees where the root was a homomer (Fig. 4B and Supplementary Fig. 4A). In the case of homomers with more than two subunits (e.g., tetra- or octamers) reversion to a monomer state is likely to need more than one steps, while a dimer can evolve from a monomer in a single step. To account for this, using the multi-state assignments, we also examined in every tree the frequency of subunit gains and losses, which shows a similar, highly significant difference as in the binary assignment (Fig. 4C and Supplementary Fig. 4B). This is also reflected in the size of the interfaces: homomers in trees with a monomer-root have much smaller (~2x) interfaces than homomers in trees where the root was also a homomer (Fig. 4D and Supplementary Fig. 4C), indicating that once large interfaces were evolved, reverting to a monomer state is much less likely than evolving a new (and relatively small) interface by a monomer. To rule out the possibility that this pattern is caused by major differences in the topologies of the trees (for example, by lower divergence from the root in trees with homomer-root), we calculated the average distance from the root (Supplementary Fig. 6A) and the average distance between the proteins (Supplementary Fig. 6B) for every tree, which show no significant differences between the two tree categories (Supplementary Fig. 6). Finally, in the case of multi-state trees with homomer roots, we also calculated whether the frequency subunit gains and losses differ; surprisingly we found no significant difference between the two (Fig. 4E and Supplementary Fig. 4D). However, this is in agreement with recent experimental findings which show that in the case of homomers that are multimers of dimers, new subunits can be gained and lost relatively easily[39,40], at least when the new interfaces are not very large or hydrophobic.

Since many of the homomers in the trees with monomer-root have small interfaces, and some of these interfaces might be crystallographic, rather than biological[6,35], we also examined whether the pattern changes if homomers with interfaces below 1000 Å² are assumed to be quaternary structure errors and are treated as monomers in the analyses. The reanalysis of the data with this extended set of monomers with ML shows that the pattern remains similar (Supplementary Fig. 7), thus, our findings are unlikely to be significantly influenced by quaternary structure errors in the PDB.

## Interface variability depends on the ancestral state of the orthogroups

Next, using the ancestral state of the roots of the multi-state analysis, we examined whether and to what degree the overlap between interface residues depends on the structural similarity of the proteins and the size of the interface. We found that within orthogroups, the overall

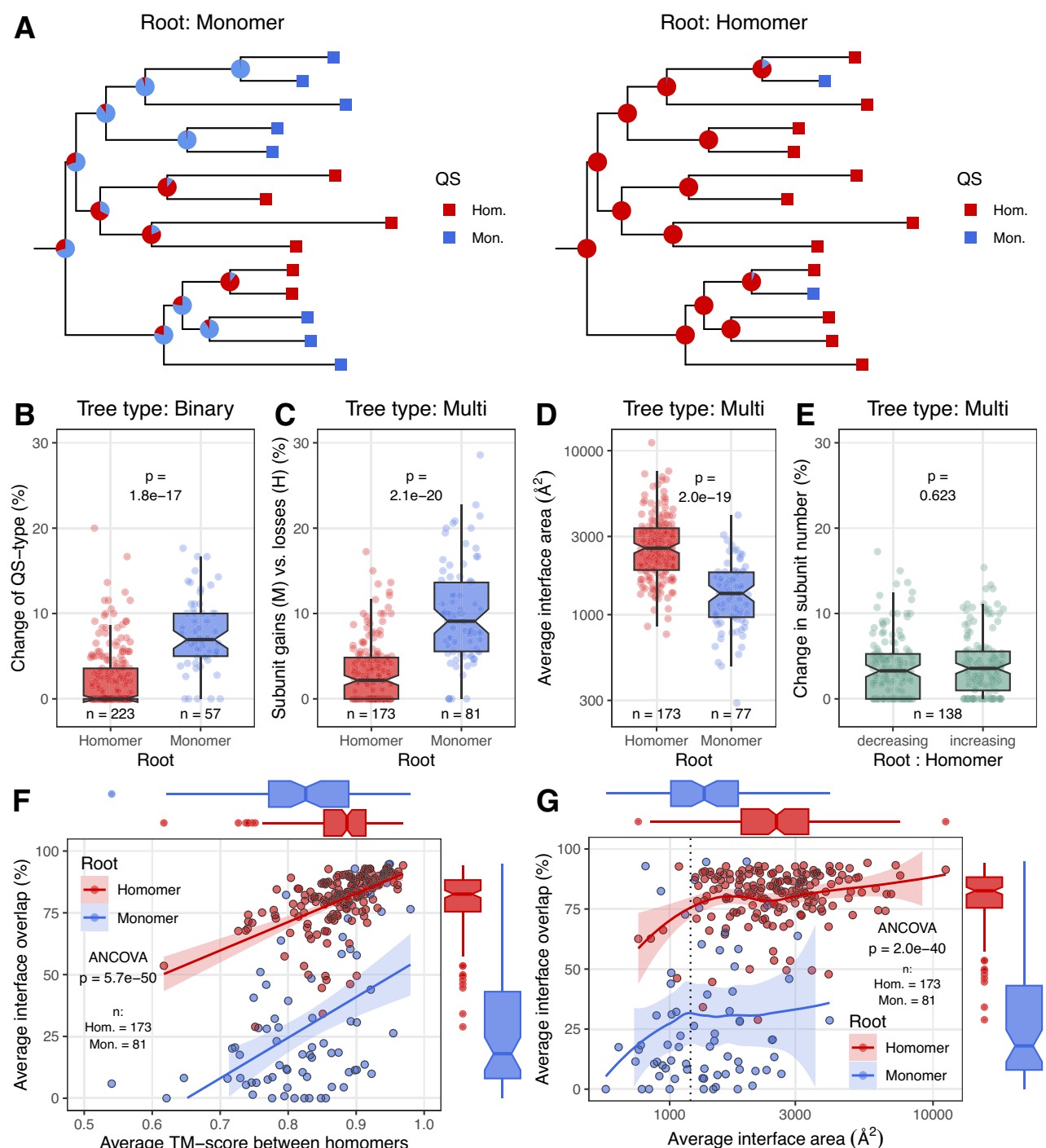

**Fig. 4 | The reversion of homomers to monomers is less likely than the evolution of homomers from monomers. A** Example phylogenetic trees illustrating ancestral states, with a monomer and homomer root. (Note that the topology of the trees is identical.) The quaternary structure of the proteins at the leaves is indicated with squares, pie charts indicate the probability of being a homomer or monomer at each node. **B** The probability of changing the quaternary structure (QS) type compared to the root is significantly higher in the trees with monomer roots than in the trees with homomer roots. **C** Similarly to the binary QS-type, the frequency of subunit losses is significantly lower in trees with homomer roots than the frequency of subunit gains in trees with monomer roots. **D** The interfaces of homomers that evolved from monomers are much smaller than the interfaces in the trees with a homomer root. **E** In the trees with homomer roots, the frequency of subunit gains and losses is similar. **F** The structural similarity of homomers is high within the

orthogroups, with TM-scores being above 0.7–0.8 for most orthogroups. However, the average interface overlap is much higher in the orthogroups with a homomer root than in the orthogroups with a monomer root. **G** Interface overlap also depends on the average size of the interfaces – below 1200 Å², it drops, suggesting that some of the small interfaces are crystallographic artefacts. The low interface overlap (< 50%) above 1200 Å² in orthogroups with monomer root indicates that new, independent interfaces evolve much more frequently in this group than in trees with homomer root. On panels (**B**–**D**) *p*-values were calculated with Wilcoxon rank-sum tests. Boxplots display the median, 25–75% interquartile range (IQR), and 1.5 * interquartile range from the hinge (whiskers). Notches are defined as 1.58 * IQR / sqrt(n). On panels (**F**) and (**G**), data points beyond the whiskers are shown as outliers.

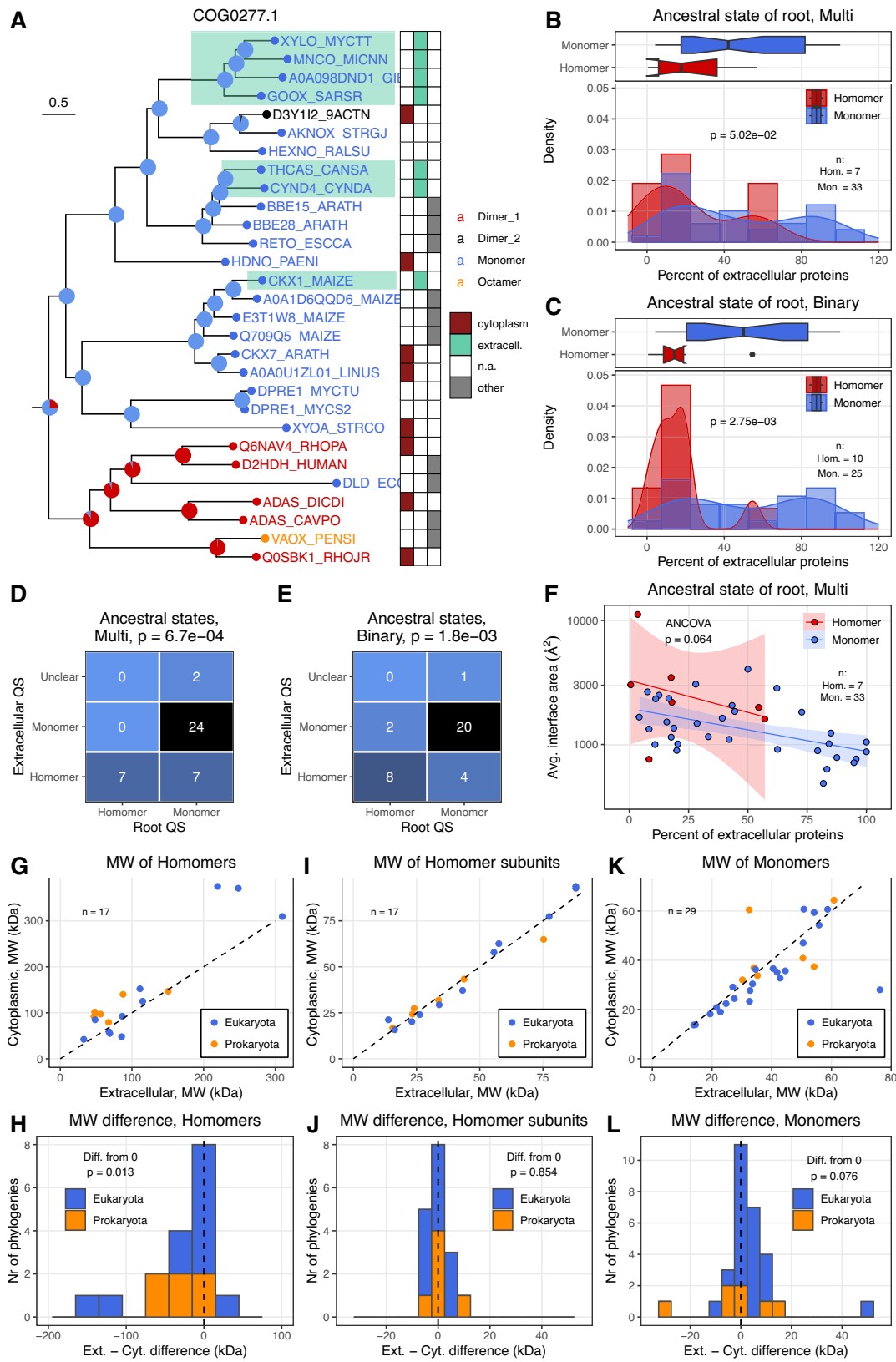

structural similarity between the subunits of different homomers is high, with the average TM-score being above 0.7–0.8 in the vast majority of the trees (Fig. 4F and Supplementary Fig. 4E). The average overlap of interfaces depends largely on the ancestral state of the tree: in the trees where the root is homomer, it is typically high, above 75%, while in the trees with monomer root, the overlap is much smaller, less

than ~50% in most cases (Fig. 4F and Supplementary Fig. 4E), indicating that in these orthogroups different interfaces evolved repeatedly. In addition, interface overlap depends on the size of the interfaces: in orthogroups where the average interface size is below ~1200 Å², the overlap drops (Fig. 4G and Supplementary Fig. 4F), irrespectively whether the root of the tree is a homomer or monomer, suggesting

**Fig. 5 | Extracellular monomers evolved from monomer ancestors. A** Example phylogenetic tree, with the quaternary structure and cellular locations of the proteins. Pie charts at the nodes indicate the probability of the number of chains for each node, extracellular proteins at the leaves are highlighted with green. **B, C** The frequency of extracellular proteins in the orthogroups, depending on the type of analysis (Binary or Multi-state). The fraction of extracellular proteins is higher in orthogroups where the ancestral state was a monomer, however, the difference is only nearly significant in the multi-state comparison due to the small number (7) of trees (Wilcoxon-rank sum tests.). Boxplots display the median, 25–75% interquartile range (IQR), and 1.5 * interquartile range from the hinge (whiskers). Notches are defined as 1.58 * IQR / sqrt(n). Datapoints beyond the whiskers are shown as outliers. *P*-values are from Wilcoxon rank sum tests. **D, E** Matrices with the quaternary structure (QS) of the root and extracellular proteins in the orthogroups which do have extracellular proteins. *P*-values (test of proportions) indicate the difference

between extracellular QS in trees with homomer and monomer roots. In the majority of cases, the QS of extracellular proteins is the same as the root, indicating that most extracellular monomers did not lose their quaternary structure but already evolved from monomers, while in the trees with homomer root most of the time, they remained homomers. This pattern is consistent with the predictions of CNE. **F** The average interface area of homomers scales similarly with the fraction of extracellular proteins in the orthogroups with homomers of monomer ancestral state. **G–J** Within orthogroups, the average molecular weight (MW) of extracellular homomers is smaller than the MW of cytoplasmic monomers, but there is no difference in their subunits (one-sample Wilcoxon-tests). This suggests that besides CNE, the selection also shapes their quaternary structure. Note that due to the larger size of eukaryotic proteins, prokaryotes and eukaryotes were treated separately, even within orthogroups. **K, L** The MW of monomers is not affected by their cellular location (one-sample Wilcoxon-tests).

that, besides rapid evolution, some of these small interfaces do contain crystallographic artefacts. However, in the case of orthogroups with monomer root, even the plateau above 1200 Å$^2$ shows only 25–50% interface overlap (Fig. 4G and Supplementary Fig. 4F), indicating that these orthogroups are characterised by the repeated evolution of homomers with new interfaces.

## Extracellular protein evolution indicates the importance of CNE

Next, we examined whether the evolution of quaternary structure in extracellular proteins is more consistent with the predictions of constructive neutral evolution (CNE, "hydrophobic ratchet"), or with the traditional view, which assumes that quaternary structure is adaptive. Extracellular proteins can be seen as a "canary in the coalmine" because the two hypotheses make different predictions: the hydrophobic ratchet predicts that once they evolve, losing interfaces is difficult, thus extracellular monomers will primarily evolve in orthogroups where the ancestral state is also a monomer. In contrast, the adaptive view is less restrictive and does not assume the persistence of quaternary structure without a fitness advantage, thus, it is more permissive for the evolution of monomers from homomers and losing interfaces.

To test this, we selected the trees/orthogroups with at least one extracellular protein, where the cellular location could be determined for at least five proteins, and the probability of the most likely ancestral state is higher than 51% (see Fig. 5A for an example). The number of such trees is limited, 35 in the binary and 40 in the multi-state case with the ML ancestral reconstruction method (Fig. 5), and 33 and 37, respectively, with MP (Supplementary Fig. 5). First, we examined the frequency of extracellular proteins, which is higher in orthogroups with a monomer root than in orthogroups with a homomer root, in the binary, and to a lesser degree in the multi-state analyses (Fig. 5B, C and Supplementary Fig. 5B, C). Next, using the most common quaternary structure of extracellular proteins in the orthogroups, we tested whether their quaternary structure changed compared to the root. We found that in the majority of trees, the most common quaternary structure of extracellular proteins is the same as the root (Fig. 5D, E and Supplementary Fig. 5D, E), and it changes compared to the root only in 22% (Fig. 5D, Multi-state analysis) and 20% (Fig. 5E, Binary analysis) of the trees (ML), and even less in the case of the MP method (Supplementary Fig. 5D, E). In the vast majority of the trees where most extracellular proteins are monomers, the root was also a monomer, and the difference between trees with homomer and monomer roots is highly significant for both analyses (Fig. 5D, E and Supplementary Fig. 5D, E). Since in some trees, the majority of proteins are extracellular (see Fig. 5B, C), which can make changes in a quaternary structure very unlikely, thus we repeated the ML analysis using only trees where the fraction of extracellular proteins is less than half of the total number of proteins (Supplementary Fig. 8). This reduced set of trees (29 and 26 for multi-state and binary) still shows a largely similar pattern, as only 27% (Supplementary Fig. 8A, Multi-state analysis) and

26.9% (Supplementary Fig. 8B, Binary analysis) of the trees indicate a change in quaternary structure. The comparison of interface sizes suggests that interface size scales similarly with the fraction of extracellular proteins irrespectively of the root, although the small number of trees with homomer root results in a very high uncertainty for the slope (Fig. 5F and Supplementary Fig. 5F). These findings indicate that reverting to a monomer state from a homomer is difficult even in the case of extracellular proteins, and most extracellular monomers evolve from monomer ancestors, in agreement with the predictions of the CNE model.

## Extracellular homomers have reduced molecular weight

In the above analysis, we focused on qualitative changes in the quaternary structure (i.e., a change from homomer to monomer). However, selection might also result in more subtle, quantitative changes, like a reduction in the size of complexes or proteins, for example, due to the better solubility of smaller complexes/monomers, or their higher diffusion rates (Fig. 2A). Diffusion rates scale with the molecular weight (MW) of proteins, and complexes, especially with many subunits, are known to have much lower diffusion rates than monomers[41]. We examined whether extracellular and cytoplasmic proteins within the same orthogroups have similar MW and found that extracellular homomers do show a statistically significant reduction in MW ($p = 0.013$, Fig. 5G, H), but not their subunits (Fig. 5I, J), or monomers (Fig. 5K, L), indicating that quaternary structure is affected by cellular location, but not protein length. (Note that Prokaryotes and Eukaryotes were treated separately, even within the same orthogroups, because Eukaryotic proteins are generally larger than Prokaryotic ones.) This suggests that selection is also likely to contribute to the loss of subunits/interfaces in extracellular proteins, although the uncertainty associated with the analysis is high due to the small number of orthogroups having both extra- and intracellular homomers.

## Amino acid synthesis cost is unlikely to be a major determinant of homomer frequency

The synthesis of amino acids is metabolically costly, and in *E. coli*, the synthesis cost, measured as the number of high energy ATP P-bonds required for synthesis in the synthesis pathway, can range between 11 P-bonds (Alanine, Glycine, Serine), to 74 (Tryptophan)[42], and hydrophobic residues with large side chains are characterised by high synthesis costs (e.g., Tryptophan, Tyrosine, Phenylalanine). Such costs can change when the entire metabolic network is considered or using different substrates[43] nevertheless, maintaining a hydrophobic interface is likely to be more costly than having an equivalent solvent-accessible (hydrophilic) surface. Thus, having unnecessary hydrophobic interfaces might be deleterious and is unlikely to be a completely neutral trait. We examined whether the synthesis costs of proteins differ in homomers and monomers of *E. coli*, and found that, as expected, the surface of subunits (that included interface residues,

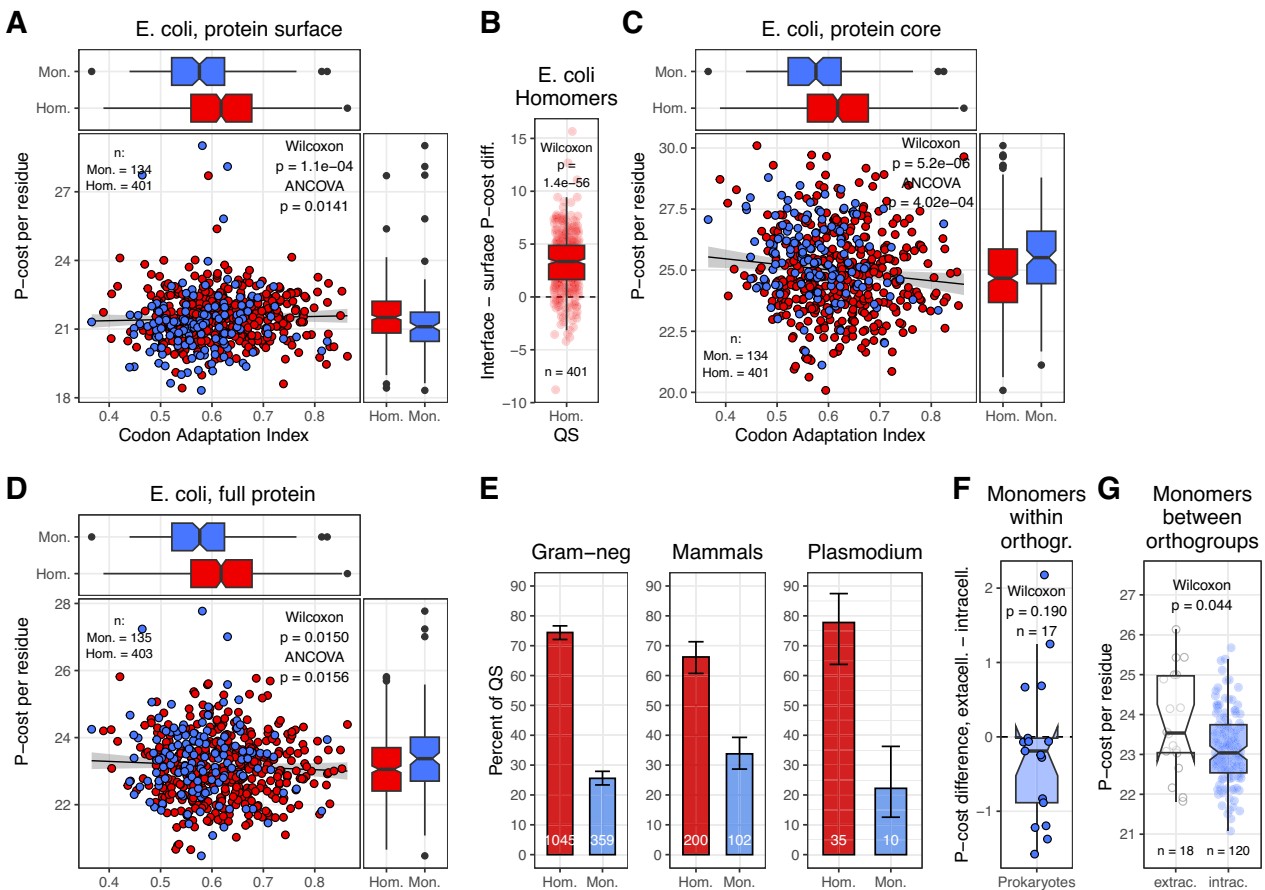

**Fig. 6 | Synthesis costs of homomers and monomers. A** Cost of surface residues (including interface) in *E. coli*. **B** Difference between interface and solvent accessible surface residues in homomers of *E. coli*. **C** Cost of core residues in *E. coli*. **D** Cost of full protein (including signal peptides) in *E. coli*. To remove redundancies, *E. coli* proteins were clustered at 30% sequence similarity, resulting in 406 homomers and 134 monomers. As expected, the per-residue synthesis cost of the subunit surfaces is somewhat more costly in homomers than in monomers due to the higher synthesis cost of interfaces, however, the synthesis cost of the core and full protein is more costly in monomers, even if codon adaptation index is included as a covariate. **E** The frequency of homomers and monomers in selected taxa (Gram-negative bacteria, mammals, *Plasmodium* [malaria parasite]) with different amino acid metabolism (numbers indicate the number of clusters). *Plasmodium* takes up most amino acids from its host, in mammals most hydrophobic amino acids are essential, while most bacteria can synthesise all amino acids. (See also Supplementary Fig. 11). **F** Within orthogroups, the per-residue synthesis cost of prokaryotic extracellular and intracellular monomers does not differ significantly. **G** Between orthogroups, the synthesis cost of extracellular monomers is not lower than the costs of monomers in orthogroups without extracellular proteins. On all panels where present, boxplots display the median, 25–75% interquartile range (IQR), and 1.5 * interquartile range from the hinge (whiskers). Notches are defined as 1.58 * IQR / sqrt(n). Datapoints beyond the whiskers are shown as outliers.

see "Methods") has a higher per-residue synthesis cost in homomers (Fig. 6A), particularly in interfaces (Fig. 6B). However, the protein core is more costly in monomers (Fig. 6C), and the total synthesis cost of proteins is also slightly higher in monomers, even when differences in codon adaptation index (which integrate expression levels) are taken into account (Fig. 6D). This raises the possibility that in *E. coli* (and species with similar metabolic pathways), the high frequency of homomers in the cytoplasm is nevertheless the result of selection against monomers for metabolic reasons, if oligomerisation enables the reduction of total synthesis costs of proteins.

This hypothesis can be largely ruled out by the examination of taxonomic groups with amino acid auxotrophies. In mammals, most hydrophobic amino acids (F, I, L, M, V, W) are essential and are taken up from the environment, and two (C, Y) are conditionally essential. Thus, there is no, or just reduced synthesis cost associated with them. Intracellular parasites like *Plasmodium sp.* or *Toxoplasma* take up most amino acids from their host[44], and parasitic bacteria like *Tenericutes* (which include *Mycoplasma*) are also auxotrophic for most amino acids[45]. In contrast, auxotrophy is rare in plants, and most bacteria (78.4%) can synthesise all amino acids[45]. Despite these major metabolic differences, the ratio of homomer and monomer enzymes in the cytoplasm does not vary dramatically between these taxa (Fig. 6E and Supplementary Fig. 11), and auxotrophic endoparasites also consistently show the same pattern, even though the number of their currently available PDB entries is very low. In addition, theory suggests[46] that in eukaryotes, the cost of protein production is usually not high enough to be perceived by natural selection, while it is in prokaryotes.

It has also been suggested that in prokaryotes, extracellular proteins, especially flagellar and fibrous proteins, have reduced synthesis costs because their amino acids are not recycled by the cell[47]. Using the proteins of the orthogroups, we examined whether this pattern is present also in enzymes (and whether it can contribute to the low frequency of homomers in extracellular proteins). The results indicate that in our dataset, the synthesis costs of extracellular monomers are not lower than the synthesis costs of intracellular monomers, neither within orthogroups (Fig. 6F) nor between orthogroups (Fig. 6G), and also when all prokaryotic extracellular monomers are compared to all prokaryotic intracellular monomers (*n* = 176 and 1426, respectively; *p* = 0.487, Wilcoxon rank sum test). Thus, location itself is unlikely to

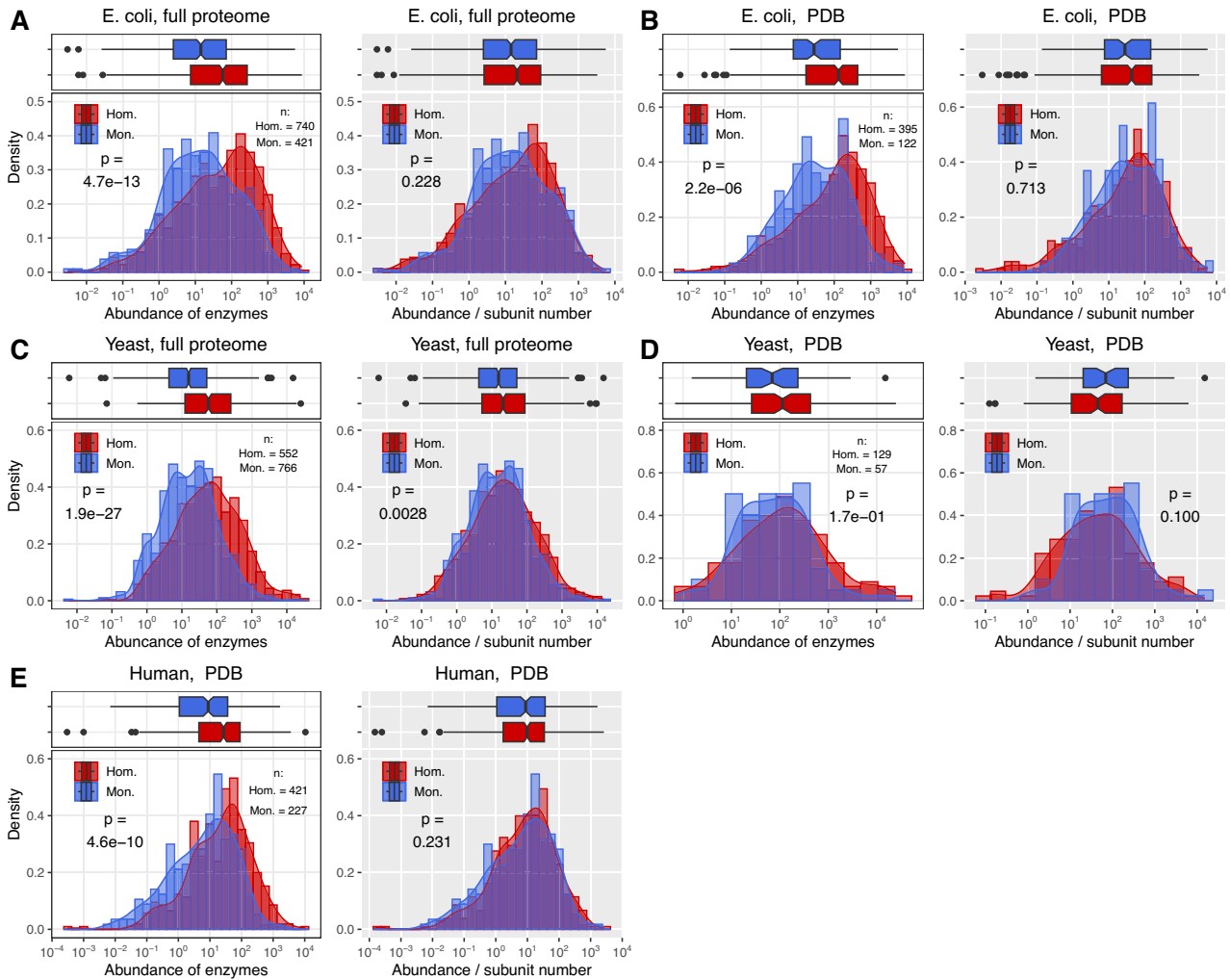

**Fig. 7 | Homomers are more abundant than monomers. A** Abundance of *E. coli* enzymes using the full proteome, which includes proteins from the Homomer atlas where oligomerisation was predicted with AlphaFold2, and also the entire PDB, including the low coverage structures. The white panel indicates raw abundances from the paxDB, and the grey panel indicates abundances scaled with the putative number of subunits in each complex, thus it is proportional to the number of particles. **B** Abundance of *E. coli* enzymes, but using only the filtered enzyme dataset, with high coverage structures in the PDB. **C, D** The same for Yeast. **E** Abundance of Human enzymes, using only the high-coverage PDB dataset. Except for panel (**D**), in all three species, homomers are more abundant than

monomers (panels with white background), but the difference disappears when the scaled abundance (-number of particles) is used. This pattern is consistent with the hypothesis that the high abundance of proteins (macromolecular crowding) facilitates the evolution of homomers. However, it also indicates that not just the number of proteins, but also the number of particles they form is regulated by the cell, suggesting that oligomerisation contributes to the maintenance of cellular homoeostasis. On all panels, boxplots display the median, 25–75% interquartile range (IQR), and 1.5 * interquartile range from the hinge (whiskers). Notches are defined as 1.58 * IQR / sqrt(n). Datapoints beyond the whiskers are shown as outliers. All *p*-values were calculated with Wilcoxon rank sum tests.

have a major effect on the synthesis cost of these enzymes, and their quaternary structure.

However, proteins that are more expressed or abundant were shown to have lower average synthesis costs[42,48,49], thus the lower overall metabolic cost of homomers in *E. coli* can also reflect higher protein abundance or expression, which is in agreement with the hypothesis that macromolecular crowding is a significant factor in the evolution of oligomers (see below).

### Homomers are more abundant in the cell than monomers

If the high frequency of oligomerisation in the cytoplasm evolves due to macromolecular crowding, that predicts that oligomers will be formed primarily by proteins that are more abundant than monomers. We tested this hypothesis using the *E. coli*, yeast and human enzymes, and abundances obtained from the paxDB database[50]. Besides using our high coverage enzyme dataset from the PDB where the quaternary structure is known, in the case of *E. coli* and yeast, both the

heteromers[51] and homomers[52] of their entire proteomes are known or predicted, thus their monomers can also be defined, and proteome scale comparisons of enzymes are possible. Our results indicate that in all three species, homomers have higher abundances than monomers (Fig. 7, white panels), the only exception being the comparison in the yeast set of the PDB (Fig. 7D), most likely due to the low numbers of available proteins. In addition, the pattern is present both in cytoplasmic (Supplementary Fig. 12) and non-cytoplasmic (Supplementary Fig. 13) proteins. This is consistent with the hypothesis that homomer interfaces evolve due to macromolecular crowding, i.e., due to the high frequency of interactions in the cell and the minimisation of excluded volume[29,30].

In addition, when the abundances are scaled down with the number of subunits of each complex (or, in the case of predicted structures, the putative number of subunits, see "Methods"), thus when the abundance of the complexes/particles they form is compared, in all three species the difference between homomers and

monomers largely disappears (Fig. 7, grey panels). This suggests that not only the abundance of enzymes (i.e., the number of catalytic sites), but also the abundance of their complexes (the number of particles they form) is regulated in the cell, the two might be shaped by separate processes, and oligomerisation allows a certain level of uncoupling between the two. Thus, besides biochemical function, the evolution of oligomers and their frequency might also be shaped by the necessity to maintain cellular homoeostasis, most likely by their effect on the properties of the cytoplasm, like viscosity/fluidity, osmotic pressure or diffusion rates.

However, several studies have demonstrated that in eukaryotes, oligomerisation increases the half-life of protein complexes and reduces their degradation rate, due to burying ubiquitinoylation sites and degrons in interfaces[18,26,53,54]. A similar interface effect has not been described in bacteria so far, nevertheless, it may exist, as interface residues are generally enriched at the C-termini of proteins[55], and may overlap with C-degrons. Thus (at least in eukaryotes and archaea), the higher abundance of oligomers may be the consequence of oligomerisation, the cause of it, or both. Mallik and Kundu demonstrated that in yeast, the half-life of homomers scales with their buried interface area[18], and using the high-coverage enzyme dataset, we examined whether the association between abundance and buried interface area is similarly strong as between abundance and oligomerisation (Fig. 8 and Supplementary Fig. S14). We found that both in *E. coli* (where the ubiquitin-proteasome pathway doesn't exist) and human datasets there is a significant positive correlation between the buried interface area of homomers and their abundance, thus the presence of interfaces and complex formation may influence abundances. However, its correlation coefficient ($R^2$) is lower than between quaternary structure and abundance (Fig. 8), especially when only dimers are used, where only *E. coli* is significant (Supplementary Fig. 14). Thus, the higher abundance of homomers is unlikely to be purely the consequence of lower degradation rates due to having interfaces. The presence of degrons in interfaces is likely to be used as a regulatory mechanism by the cell, however, the fact that interfaces do influence protein abundance through degradation, together with macromolecular crowding, can result in a positive feedback between the two: abundant proteins oligomerise, which in turn reduces their degradation rate, resulting in even higher abundances and interaction frequencies. We hypothesise that besides the hydrophobic ratchet, this "degradation ratchet" might be an additional process causing the accumulation of interfaces over evolutionary timescales, at least in eukaryotes.

## Discussion

Our results indicate that in extracellular environments, most enzymes do not form homomers (Fig. 2), indicating that the factors responsible for the evolution of oligomers, both the adaptive and neutral ones, are primarily present in the intracellular environment. Even though hydrophobic interfaces are likely to have a metabolic cost (Fig. 6), the general pattern of homomer evolution in enzymes is consistent with the predictions of CNE ("hydrophobic ratchet"): the transition from monomers to homomers is easier than the reverse process (Fig. 4), extracellular monomers evolved primarily from proteins that were monomers also in their ancestral state, and in the majority of cases their evolution did not involve the loss of interfaces (Fig. 5 and Supplementary Fig. 8). The bias in interface evolution also explains previous observations that the order of the assembly of protein complex subunits reflects their evolution[14,56,57], if new interfaces are added as the complex evolves, but old interfaces are rarely lost. In addition, our analyses indicate that other factors, like differences in catalytic constants (Supplementary notes, Supplementary Figs. 9 and 10) or differences in the synthesis cost of amino acids, are unlikely to explain the differences in homomer frequencies between the extracellular and cellular environment.

The high frequency of hydrolases among extracellular enzymes (Fig. 3) suggests that interactions with the solvent might be among the key factors responsible for their rare oligomerisation. Oligomerisation might be deleterious in the extracellular space due to the exposed hydrophobic interfaces if homomers dissociate due to the low concentration of their subunits, or monomers can even be adaptive due to their smaller size if selection favours high diffusion rates (Fig. 2A).

In many enzymes, the contribution of quaternary structure to their function is beyond doubt (e.g., allosteric oligomers). However, enzyme oligomerisation is environment-dependent (Fig. 2), while biochemical function is not; in some cases, oligomerisation is known to be gratuitous[5,15], and the interfaces of homomers with identical functions are not always conserved (see Fig. 1). This indicates that some of the factors that facilitate the evolution of oligomers are not directly related to the biochemical function of the protein, but rather to the characteristics of the cellular environment. One such characteristic is likely to be the macromolecular crowding of the cytoplasm, which can result in high interaction frequencies, facilitating the evolution of interfaces. However, oligomerisation may also be related to the maintenance of cellular homoeostasis, like water availability[19], and proteostasis[15,18,26], determining the viscosity/fluidity of the cytoplasm and diffusion rate of proteins within the cell. The higher frequency (Fig. 2) and abundance (Fig. 7) of homomers than monomers, and the fact that the difference in abundance disappears when it is scaled with subunit number is consistent with both possibilities (macromolecular crowding and cellular homoeostasis). The evolution of complexes (e.g., dimers) may simply allow the uncoupling between biochemical and biophysical constraints of the cell, for example, if the cell needs N binding sites for catalysis, but only N/2 particles for the optimal cytoplasmic fluidity/viscosity, osmotic pressure or diffusion rate (these parameters are usually correlated). Due to excluded volume effects, complex formation also allows a higher number of proteins (and denser cytoplasm) in the same cell volume than with monomers, explaining the presence of homomers in viruses. A process like this is easy to reconcile with the neutral pattern of interface evolution, even if they have a metabolic cost, as from the perspective of maintaining the correct cytoplasmic fluidity, it may not matter which proteins form oligomers. Thus, interfaces of individual proteins can vary and evolve effectively randomly, as long as selection maintains the biophysical properties (and oligomer frequencies) of the cell within the physiological range. This would also explain the relatively constant ratio of homomer and monomer enzymes in different taxonomic groups (Supplementary Fig. 11). In addition, the weak, nevertheless positive correlation between interface size and protein abundance suggests that besides the hydrophobic ratchet, a "degradation ratchet" may also contribute to the accumulation of interfaces over evolutionary timescales.

The relative contribution of oligomerisation to these processes remains to be seen, and an important limitation of our work is that it does not take into account heteromers, due to their very incomplete annotation. Healthy cells maintain a narrow size range[58], and the dilution of the cytoplasm due to cellular overgrowth results in impaired cell function[59]. Recent work indicates that protein condensates play a role in the regulation of water potential in the cell[19] and osmotic pressure[60,61], thus, the frequency and abundance of oligomers in the cytoplasm may help to establish the "default" amounts of available water. The lower diffusion rate of oligomers might also allow a certain level of compartmentalisation within the cytoplasm[15,41] and, in consequence substrate channelling (and enzyme assemblies)[62], which have been observed for several metabolic pathways[62]. In contrast, in the extracellular space, maintaining enzyme assemblies is probably not possible, and selection might favour the highest diffusion rate. In the cytoplasm, diffusion rates show considerable spatial variability[63], and several studies have demonstrated that the diffusion rates of proteins are also highly

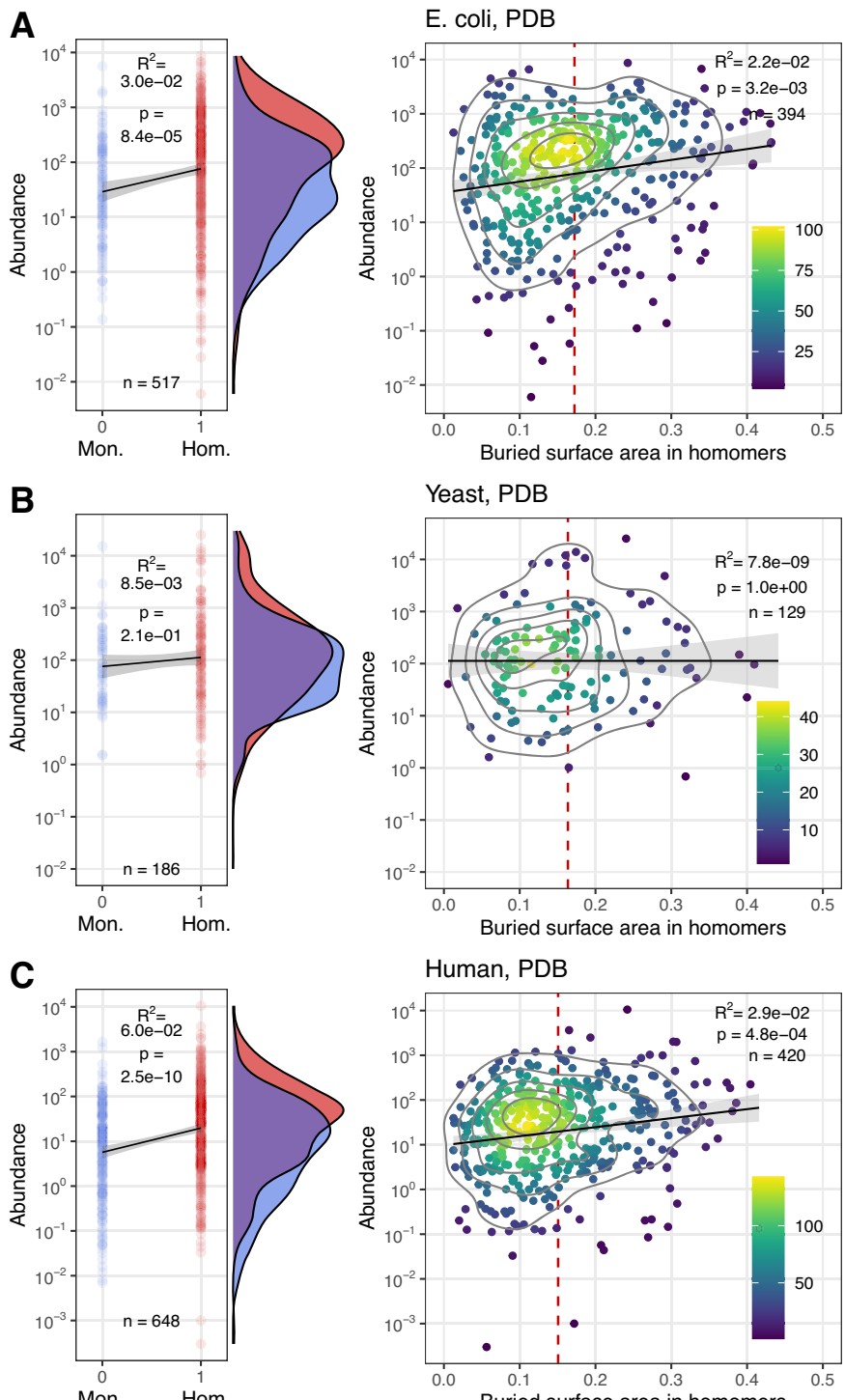

**Fig. 8 | The higher abundance of homomers is unlikely to be the by-product of their lower degradation rates due to having interfaces. A** E. coli. **B** Yeast. **C** Human. The left panels indicate the correlation between quaternary structure and abundance and the corresponding abundance density plots, the right panels indicate the correlation between abundance and the relative surface area of homomers that are buried in the interface. *p*-values were obtained with the Pearson correlation test. Red vertical lines indicate the average buried surfaced area. While R² is low in all cases, it is significant in *E. coli* and Humans, and in both cases the association between buried surface area and abundance is weaker than between quaternary structure and abundance. This indicates that the positive correlation reported previously in eukaryotes between relative interface size and protein half-life is not sufficient to explain the abundance difference between homomers and monomers. (See also Supplementary Fig. 14, showing only dimers).

variable and depend both on the size of oligomers[41,64,65], as well as the surface characteristics and charge of proteins[66,67] (which are influenced by their interfaces). However, in *E. coli*, even for large proteins (582 kDa), the cytoplasmic diffusion rate remains high enough to traverse the cell several times every minute[64], and the typical homodimer is much smaller than that. In addition, the frequency of homomers in enzymes does not appear to differ dramatically in taxonomic groups that face different osmotic pressures[20], have

different cell walls/membranes, and are characterised by proteins with different surface charge[68], like Gram-negative vs. Gram-positive bacteria, Plasmodium species or mammals (Supplementary Fig. 11), and even in the small number ($n = 14$) of enzymes from halophilic bacteria there is only a single monomer. This indicates that on evolutionary timescales, proteomes adapt to environments with different osmotic pressures, primarily with protein surface charge, and that oligomerisation evolves largely independently from it. However, if oligomerisation influences the fluidity of the cytoplasm, that is likely to result in the observed relatively constant ratios between homomers and monomers (Supplementary Fig. 11).

Finally, our results also have practical implications and suggest that in industrial applications where the conditions are comparable to the extracellular environment rather than the cytoplasm, engineering monomeric forms of certain enzymes might be advantageous compared to their native oligomeric form, either due to being less aggregation-prone, or lower synthesis costs.

## Methods

### Data sources

We downloaded all proteins from the Protein Data Bank (PDB) with homomer or monomer quaternary structure, resolution better than 3 Å, and where the PDB structure contains a minimum of 80% of the residues of the UniProt sequence. Only proteins with a minimum length of 100 amino acids were used; when signal peptides and transit peptides were present in the UniProt annotation these were not included in the 100 amino acids. For every UniProt entry, we used all of its PDB structures to identify the quaternary structure of the protein, i.e., in the case of monomers, all first biological units are monomers, and in the case of homomers, at least one of the PDB entries form a homomer, and none are part of a heteromer. PDB entries with multiple biounits, where the biounits have variable quaternary structures (e.g., some are monomers, others are homomers), were not used. In addition, sequences annotated as antibodies and MHC proteins, PDB entries with chimeric sequences, proteins forming fibrils, and virus proteins were also not included in the analyses.

The enzymatic activity of proteins (EC numbers) was extracted from UniProt. For proteins without an EC number in the UniProt, enzyme function (EC number) was predicted with the CLEAN tool (v.1.0.1)[69]. For the CLEAN predictions, we used a minimum distance of 0.5 as the cutoff for enzymatic activity: below 0.5 the protein was not considered an enzyme. Altogether we identified 3740 eukaryotic, and 8228 prokaryotic sequences that meet the above criteria.

The cellular location of proteins was determined using the PSORTdb database[32] (v4.0) and PSORTb tool (v3.0) for prokaryotes and with DeepLoc v2.0[31] for eukaryotes. We used the full sequences (containing signal and transit peptides) to determine the cellular location. When a sequence was annotated as both extracellular and cytoplasmic or also having other cellular locations, the ones having a Sec/SPI or Tat/SPI signal peptides were classified as extracellular proteins. Signal peptides were identified with the SingnalP6 tool[70]. Proteins that are neither cytoplasmic nor extracellular, or are present both in the cytoplasm and other organelles (but are not extracellular) were annotated as "other".

Catalytic constants ($K_M$, and $k_{cat}/K_M$) of proteins were downloaded from the BRENDA database[71], the constants of protein mutants were not used. We used the lowest $K_M$ and the highest $k_{cat}/K_M$ for all substrates (Supplementary Fig. 9) and the natural substrates (Supplementary Fig. 10) of the proteins separately, because for a large number of proteins where catalytic constants are available, the constants for the natural substrates are not known.

### Clustering, ortholog identification and phylogenetic analyses

To remove redundancies, we clustered the sequences at 30% sequence similarity and 90% coverage with the MMseqs2 tool[33] (using the flags: --alignment-mode 3 --min-seq-id 0.3 -c 0.9 --cov-mode 1 --cluster-mode

2-s 8.5). Eukaryotic, Prokaryotic, and taxon-specific (Supplementary Fig. 11) sequences were clustered separately.

To identify orthologous proteins in the pooled Eukaryotic and Prokaryotic sequences (11968 sequences), we used the web version of eggNOG-mapper[38] with default settings, except for realigning the queries to the entire PFAM database. The lowest root-level orthogroups were used for every protein. In addition, we required that in each orthogroup, the sequences have shared PFAM domains, and orthogroups with less than 10 sequences were not used in the downstream analyses. Altogether, we identified 311 orthogroups, of which 108 had 20 or more sequences, and 203 had 10–19 sequences.

Within orthogroups, the quaternary structure was determined for every protein, and the overlap between interface residues was calculated (see below) for every possible protein pair. Subgroups within homomers were identified by identifying the connected components with homologous interfaces, i.e., networks of proteins where in every protein pair, one protein had at least 50% of its interface residues structurally aligned with the interface residues of the other protein.

Multiple sequence alignments of the orthogroups were made with MAFFT-DASH (v7.520)[72], with the L-INS-I method (--maxiterate 1000 --localpair). This integrates sequence and structure alignment and improves the quality of alignments of highly diverged sequences if their structures have similar folds. Phylogenetic trees were made with IQ-Tree v.2[73] and were subsequently rooted with the Minimal Ancestor Deviation method (MAD v2.2)[74]. Ancestral states (probabilities) of the tree nodes, including the root, were determined with the ace function of APE v.5.7 (Analyses of Phylogenetics and Evolution) library of R[75], using the discrete maximum likelihood method, and the equal rates (ER) model. For maximum parsimony, we used the ancestral.pars function of the Phangorn v.2.10 R library[76], with the ACCTRAN model. Of the 311 phylogenetic trees, only those were used where the most likely ancestral state of the root had at least 51% probability. The frequency of changes in subunit numbers (or quaternary structure) was calculated as the number of parent-child node-pairs in the tree where the subunit number (quaternary structure) was different in the ancestral and child node, divided by the total number of node-pairs of the tree where the quaternary structure could be determined (or was known, i.e., the leaves of the tree) for both nodes.

### Identification of surface and interface residues, and their conservation

The solvent-accessible surface area (SASA) of all residues in the protein complexes was determined with FreeSASA v1.1[77], in all complexes, their subunits, and monomers. The size of the interfaces was defined as the total difference in the SASA between the homomers and the sum of the SASAs of their subunits, divided by the number of subunits in the complex. The relative buried surface was defined as the difference between the SASAs of all subunits minus the SASA of the assembled complex, divided by the sum of the SASAs of all subunits. Interface residues used in the analyses of conservation were defined as residues with different SASA in the subunits and the full homomer, if the difference was larger than 10% of their SASA in the subunit, and the relative solvent accessibility in the subunits was higher than 20% (using the full amino acid areas defined by Miller et al.[78]). Surface residues were defined as residues which are not part of the interface and have relative solvent accessibility above 20%. The hydrophobicity of interfaces and surfaces was measured as the fraction of C, F, I, L, M, V, and W residues in them.

The conservation of residues was determined using the ConSurf database[34]. We downloaded the summary files for every structure and calculated the difference between the average conservation of the interface and surface residues using the normalised conservation score.

Molecular weight (MW) was calculated as the sum of the MW of all residues of the UniProt sequence (minus waters, and excluding the

signal and transit peptide, when present), which in the case of complexes was multiplied by the number of subunits.

### Gene Ontology analyses

Gene Ontology (GO) annotations for all sequences were downloaded from UniProt; the full ontology network (go.obo file) from https://geneontology.org. As the completeness and depth of the annotations are highly variable for different proteins, for every UniProt GO term, we also identified all of their parental terms by traversing the "is_a" tags of the go.obo file to the highest level Molecular Function or Biological Process terms. Next, using the full lists of GO terms (UniProt + parental), we determined the GO-term enrichment of Molecular Function and Biological Process terms with GeneMerge (v1.4)[79]. The analysis was performed separately for Eukaryotic and Prokaryotic proteins.

### Calculation of interface overlap

For each homomer in the orthogroups, interface residues were determined as described above. Next, using TM-align (v. 20190822)[80], we made all possible structural alignments between the first subunits of the complexes, and in the TM-align output, we identified the residue pairs that correspond to each other in the two structures. Interface overlap was calculated as the number of structurally aligned residue pairs where both residues are part of an interface, divided by the size of the smaller interface. (Similarly, from the two TM-scores of the structural alignment, the one normalized by the smaller structure was used). Average overlaps (and also average TM-scores) were calculated as the average of all possible pairwise comparisons in the orthogroup.

### Calculation of protein synthesis cost

Synthesis costs of individual amino acids were obtained from Akashi and Gojobori[42]. The codon adaptation index (CAI) for each *E.coli* gene (taxids 562 and 511145) was obtained from the Codon Statistics Database[81]. ATP costs of amino acid synthesis from Kaleta et al.[43] (glucose medium) resulted in clear positive correlations between CAI and per-residue ATP cost, and were not used. The protein surface was defined as the residues with relative solvent accessibility larger than 0.2 in the monomer subunits (that includes also the interface residues), the remaining residues (excluding signal peptides) were used as the core. The per-residue cost of the full protein included also the signal peptides, when they are present. In the analyses of proteins in orthogroups (Fig. 6F, G), we used sequences without signal peptides, and for Fig. 6F, we also used orthogroups with less than 10 sequences, to increase the power of the test.

### Identification of quaternary structure in proteomes, and protein abundance calculation

Protein abundance data for the E.coli, Yeast and Human proteomes were downloaded from the paxDB database[50]; for each species, the integrated dataset for the whole organism was used. For *E. coli* and Yeast, homomers were identified as the proteins that either have a predicted AlphaFold2 structure in the recently published Homomer atlas[52] or a homomer entry in the PDB (irrespectively of its coverage of the protein sequence, thus incomplete structures were also used). Proteins that form a heteromer in the ComplexPortal[51] or the PDB were excluded. Monomers were defined as proteins that are neither homomers nor heteromers (using both PDB, Complex Portal and predicted homomers). Only enzymes (i.e., proteins having an EC number) were used in the analyses. In the case of Human data, proteome scale comparisons are not yet possible, due to the incomplete annotation of complex-forming proteins, thus we used only the enzymes having a high coverage structure in the PDB (see the "Data sources" section above). In the case of predicted homomers, the number of their subunits was estimated by the number of subunits of the closest homologue in the PDB, as provided by Schweke et al.[52];

when such data was not available, the homomers were assumed to be dimers.

### Statistics and data visualisation

All statistical tests were two-sided, and were performed with R (v.4.1.2). Data visualisations were performed with the ggplot2 (v.3.3.5) R package. Boxplots are displayed as defined by the geom_boxplot function of ggplot2, with median, 25–75% interquartile range (IQR) for the box, and up to 1.5 * interquartile range from the hinge for whiskers. Notches, when present, are defined as 1.58 * IQR / sqrt(n). Data points beyond whiskers are considered outliers and are displayed when the underlying raw data is not shown on the plot. On every panel of the figures where more than one statistical test was performed (i.e., more than one p-value is provided), corrections for multiple testing were performed with the Benjamini-Hochberg method, except for the Gene Ontology analysis (Fig. 3E), where the Bonferroni correction was used.

### Reporting summary

Further information on research design is available in the Nature Portfolio Reporting Summary linked to this article.

## Data availability

Data necessary to reproduce all figures, Supplementary Figures, and tables is available without restrictions at Zenodo (https://doi.org/10.5281/zenodo.11410391). Source Data is provided as a Source Data file. Source data are provided in this paper.

## Code availability

Code to reproduce all figures, Supplementary Figures, and tables is available without restrictions at Zenodo (https://doi.org/10.5281/zenodo.11410391).

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

## Acknowledgements

We thank Xiaozhi Fu, Joseph Marsh and Roberto Steiner for critically reading the manuscript. The study was supported by King's College London (Starting grant for A.Z.) and Swedish Research Council (Vetenskapsrådet) starting grant no. 2019-05356 (A.Z.), Formas early-career research grant 2019-01403 (A.Z.) and Marius Jakulis Jason Foundation (A.Z.). This work used the resources of the Computational Research, Engineering and Technology Environment (CREATE) of King's College London.

## Author contributions

G.A. conceived the project, designed and performed the analyses, and wrote the first version of the manuscript. A.Z. contributed to the analysis design and read and edited the manuscript.

## Competing interests

The authors declare no competing interests.
