## [Peer Review File · Nature Communications]

Cellular location shapes quaternary structure of enzymesREVIEWER COMMENTS

Reviewer #1 (Remarks to the Author):

This paper provides a substantial phylogenetic analysis of gains and losses of multimeric features of proteins in both prokaryotes and eukaryotes. Reaching the conclusion that the data are supportive of the common neutral evolution of homomers, as suggested previously by Lynch. They do not entirely rule out some selective aspects, as intracellular proteins behave differently from extracellular proteins, which plausibly has to do with the encounter rates of appropriate partners in these different cellular environments.

A useful aspect of the paper is the placement of the analyses in a phylogenetic context, as the authors attempt to quantify gains vs. losses on phylogenies. They conclude that gains are more common than losses, and that this results from constructive neutral evolution operating in a ratchet-like fashion. I think some more care in the analysis and/or further discussion of the implications would be useful here.

Although those who speak of CNE often invoke the ratchet metaphor, this actually implies a stronger statement about evolution, i.e., that it is proceeding in a directional manner with the final equilibrium state being all multimers. However, if the strength of interfaces can slowly walk forward in a stepwise fashion, then there is no obvious reason why they cannot walk backwards. At equilibrium, averaging over an entire phylogenetic tree, there then ought to be equal numbers of gains and losses. The two rates might be different, but at equilibrium, the most common type would be the one with the lowest ratio of gain to loss rates, and vice versa.

Thus, if the authors are correct in inferring much higher numbers of gains than losses, the protein world must not be in evolutionary equilibrium, which would be an interesting point itself. For this reason, I think some care needs to be taken in this analysis. First, to obtain unbiased estimates, it seems that the authors should include trees with no transitions to obtain less biased estimates of overall rates and states.

Second, some further scrutiny needs to be given to the matter of switches in interfaces. The authors do give this matter some attention, but it is not clear that it has entered into the phylogenetic analyses. If, for example, a subclade is viewed as having gained (at its base) a multimeric state, it would be illuminating to know whether all members of the subclade are using the same interface. If they are not, one interpretation would be that there have been losses (unrevealed by coarse-grained phylogenies) followed by regains.

Depending on how all this turns out, some further discussion of the ratchet view of evolution might be warranted.

Finally, a popular model for the evolution of multimers invokes a domain-swapping scenario,

whereby a monomer with an internal pair of binding domains loses the hinge (by deletion) and then must swap domains with a partner to close the loop. If the data allow the identification of such multimers, it might be worth taking a closer look at the data for prokaryotes vs. eukaryotes, as Lynch has pointed out the potential barrier to dimerization by this route in diploid eukaryotes resulting from selection against heterozygotes.

Reviewer #2 (Remarks to the Author):

Abrusán and Zelezniak demonstrated differences in homomer versus monomer binding interfaces, with the several conclusions on how cellular location drives the evolvability of enzyme quaternary structure. In general, the manuscript is written with great clarity and contains a conclusive and well-reasoned analysis of PDB structures showing there is an observable difference in monomers versus homomers. However, beyond the thorough analyses, I struggled to understand what the conceptual advancement was, as much of the discussion appears as rehashing literature already cited in the introduction without specific connection to the presented results. The manuscript would also benefit from less technical terms or at least a better introduction of those terms in order to be accessible for a broad readership.

Two general points

- The introduction may actually partly be based on your results, hence why the introduction and discussion seem similar. You reference Figure 2 in the introduction on line 121, the discussion on lines 354-359 seems similar to lines 81-86. The manuscript may be more clear if you state what is currently known in the field without referencing any of your findings, and then more clearly indicating how your results specifically support or reject hypotheses on what could cause location-dependent evolution (e.g., the diffusion rate but not the Michaelis-Menten analysis from the supplement appear selected on). Lines 361-365 just state existing ideas without connecting to your results, and again blurs what is introduction and what is discussion. Lines 345 and 366 seem to justify an observation if selection favours high diffusion rate and then later state that selection remains to be seen, again making it hard to know what exactly I should take away from this paper.
- I find the multistate versus binary tree analysis overly complicated, given the results are almost similar. Figure 4E looks interesting as a use of the multistate tree, but I find this hard to reconcile with the other results if losing an interface could be evolutionarily advantageous if lower MW/faster diffusion is favoured etc. Unless there is a more concrete point you want to make for differentiating homomer unit size, I would consider simplifying the numerous presented analyses by only using binary trees.

I have several more specific comments below:

- I believe the figures are cited out of order (Fig 1 is first referenced on line 259)

- Lines 106 and 117: <https://journals.aps.org/pre/abstract/10.1103/PhysRevE.98.022113> demonstrates the likelihood of aggregation (unbound) or destabilisation (nondeterministic) behaviour in a protein quaternary structure model.
- Line 96: “whether quaternary structure influences the evolution of new functions” – I’m not sure this point was addressed, instead just compared two types
- Line 148, Typo in “Figures 2 ad 3” should be “and”
- Line 156/168/170: there appears to be an inconsistent use of significant figures, unless line 156 and 157 are meant to indicate approximations
- Line 138, EC should be defined more explicitly for a general audience (as is almost done on line 390)
- Line 245 and 248: Should these refer to 4D and 4E instead?
- Line 265: What is the implication here? That complexes with monomer roots are more likely to independently evolve interacting interfaces?
- Lines 307-322: The sample sizes here are likely too small to justify this analysis. Although you caveat the uncertainty, the results (especially in Figure 5G-L) could support the opposite interpretation if the outliers were removed (or is there any evidence hinting as to why those specific homomers evolved different sizes in the cellular locations?).
- Line 335: Could consider citing <https://doi.org/10.1371/journal.pcbi.1006886>, which also discusses the order subunits reflecting order of evolution, as well as the neutral evolution of interfaces, albeit in a toy model.
- Line 365: no need for the comma before the parentheses
- Figure 4 E: typo on “decreasig”
- Figure 4 F&G: all other p-values are given exactly, whereas these give the default reporting threshold from R (2.2e-16)
- Figure 5 H-L: Does the p-value refer to both groups combined or are they treated separately (line 318).
- Line 381, I initially took this as signal and transit peptides were excluded from all analyses, yet they repeated appear in later methods (e.g., lines 398, 399, 447, etc). If this means you excluded signal and transit peptides from the 100 aa filtering, could you rephrase?

Reviewer #2 (Remarks on code availability):

The code availability statement in the manuscript is “... freely available at Zenodo (<https://zenodo.org/records/...>)”, and no code was provided to review, so I have not reviewed any, although it does not appear to be an important contribution of this paper.

Reviewer #3 (Remarks to the Author):

This paper investigates what factors influence quaternary structure in proteins. In particular, it uses bioinformatic analyses to test if there are signals for a functional benefit of oligmerization. In general, I found this paper very interesting and quite well presented and of general interest.

I have a few comments. In general, one needs to remember that selection is hard to prove in the best cases, and the absence of it even more difficult to prove. The authors are already quite careful, but they might consider making the limitations of such large scale studies more explicit. Even very small selection coefficients can in principle construct and maintain interactions especially in prokaryotes. I think it's important to bear this in mind.

My bigger concern are the ancestral state reconstructions. I object to two methodological choices. The trees are rooted with maximum ancestral deviation, which is a very inaccurate rooting technique, because it can be influenced by taxon sampling. Unfortunately, there is no free lunch when it comes to tree rooting - it is often the most difficult problem in prokaryotic phylogenies. At the very least, I would want this rooting method validated with more reliable methods (paralogy rooting, rooting according to the species tree, or DTL rooting) for some appreciable number of trees. I also found it hard to visually verify whether the roots of the trees in the main text are plausible because the trees not labelled sufficiently (not at all in Figure 4 and with hard to interpret taxon abbreviations in Figure 5).

This brings me to the biggest problem: The paper estimates ancestral states at the roots of their phylogenetic trees using a likelihood method. I'm afraid this is indefensible. The root position along the root branch is not known, but usually arbitrarily assigned by tree analysis software (often in the middle of the branch). A maximum likelihood estimate uses this position, even though there is no data to determine it. This is a very unfortunate and frequent issue in papers that use this method and it simply should not be done. The problem disappears when using parsimony, because parsimony does not use the branch lengths to estimate ancestral states. Unfortunately I cannot sign off on maximum likelihood root reconstructions.

For me to recommend publication of this very interesting and valuable contribution the issues with rooting and ancestral state estimation will have to be resolved.

Reviewer #1 (Remarks to the Author):

This paper provides a substantial phylogenetic analysis of gains and losses of multimeric features of proteins in both prokaryotes and eukaryotes. Reaching the conclusion that the data are supportive of the common neutral evolution of homomers, as suggested previously by Lynch. They do not entirely rule out some selective aspects, as intracellular proteins behave differently from extracellular proteins, which plausibly has to do with the encounter rates of appropriate partners in these different cellular environments.

A useful aspect of the paper is the placement of the analyses in a phylogenetic context, as the authors attempt to quantify gains vs. losses on phylogenies. They conclude that gains are more common than losses, and that this results from constructive neutral evolution operating in a ratchet-like fashion. I think some more care in the analysis and/or further discussion of the implications would be useful here.

Although those who speak of CNE often invoke the ratchet metaphor, this actually implies a stronger statement about evolution, i.e., that it is proceeding in a directional manner with the final equilibrium state being all multimers. However, if the strength of interfaces can slowly walk forward in a stepwise fashion, then there is no obvious reason why they cannot walk backwards. At equilibrium, averaging over an entire phylogenetic tree, there then ought to be equal numbers of gains and losses. The two rates might be different, but at equilibrium, the most common type would be the one with the lowest ratio of gain to loss rates, and vice versa.

We would like to add here one implication of the ratchet: complexes can gain interfaces both due to a neutral process (CNE) or selection, but the loss of interfaces is likely to be primarily the result of selection, because, as pointed out by Hochberg et al. 2020 Nature, neutral mutation rates drive amino acid composition to an interface-like state with a high frequency of hydrophobic amino acids.

Thus, if the authors are correct in inferring much higher numbers of gains than losses, the protein world must not be in evolutionary equilibrium, which would be an interesting point itself. For this reason, I think some care needs to be taken in this analysis. First, to obtain unbiased estimates, it seems that the authors should include trees with no transitions to obtain less biased estimates of overall rates and states.

We are not entirely sure whether this is what reviewer means, but in the analyses of Figure 4 we did use the trees where all proteins had the same quaternary structure, so there is no quaternary structure change (transition) in the tree. Reviewer has highlighted a very interesting aspect of the analysis, however, there are at least two important characteristics of the data that complicate it.

First, although the phylogenetic trees contain only orthologous proteins, in many trees these orthologs can be very distant ones, because the PDB is enriched in medically/industrially important proteins, and these are frequently ancient. They have similar folds, with TM-score usually being above 0.75-0.8, but the sequence similarity of the most distantly related proteins in the orthogroups can be very low, below 20%. Frequently, the last common ancestor (root) of these proteins had to originate before the prokaryote/eukaryote split because the orthogroups can include proteins from prokaryotes and eukaryotes, and also from eukaryotic organelles (see Figure 5A for an example). Thus, we think we cannot say that currently the protein world is not in equilibrium, only that it probably wasn't in equilibrium during the timespan that was analysed, but for many orthogroups (243 out of 311) that can mean a period longer than the history of eukaryotes because they contain proteins from more than one kingdom. To estimate whether there is currently an equilibrium between the frequency of homomers and monomers, one would need to use much shallower phylogenies or a different method. (We think testing this will be possible when a

reliable method for predicting protein quaternary structure will be available, ideally with an approach that does not rely on evolutionary information. This might be within reach at some point in the not-too-distant future, as AlphaFold3 was just released.)

Second, equilibrium depends also on the number of trees, not only the rates. So, if there are 100 trees with homomer roots where the “rate” of subunit loss is ~5%, and there are only 25 trees with monomer root where the “rate” of subunit gain is ~20%, then they might be in an equilibrium, in the sense that the overall numbers of homomers and monomers may remain stable (assuming that the trees are of the same size). But if there are 200 trees with homomer roots then the total number of interface losses is higher than gains, despite the higher rate of interface gains in the trees with monomer roots. So as the number of homomers grows in the proteomes, the easier it is to reach an equilibrium (assuming that homomers do not acquire subunits indefinitely, which is reasonable). However, our goal was less ambitious, we did not try to get an unbiased estimate of the total gains and losses of interfaces – i.e. equilibrium – across the protein world, but we wanted to test whether interfaces were generally gained easier than lost during evolution (as it is predicted by CNE), and we think our analysis allows that.

Second, some further scrutiny needs to be given to the matter of switches in interfaces. The authors do give this matter some attention, but it is not clear that it has entered into the phylogenetic analyses. If, for example, a subclade is viewed as having gained (at its base) a multimeric state, it would be illuminating to know whether all members of the subclade are using the same interface. If they are not, one interpretation would be that there have been losses (unrevealed by coarse-grained phylogenies) followed by regains.

Reviewer is correct on this, in the previous version of the manuscript, we did not distinguish between homomers with the same number of subunits but non-homologous interfaces. In the binary analysis, this is not possible, but it can be incorporated into the multi-state analysis. We did it, and it does not change any of the conclusions of the manuscript. (See the updated Figures 4 and 5 below). In the revised manuscript, we use these new figures, which differ in a few things from the previous version: on Figure 4 interface areas (Fig 4D) are given for the multi-state analysis now (previously, it was the binary analysis), and Figures 4F and 4G are now also based on the multi-state analysis and not the binary one. Neither of these changes results in qualitative differences in the patterns we observe, or changes in the conclusions of the work. On Figure 5A, protein D3Y1I2_9ACTN is a dimer, but it has a different interface than the other homomers on the figure, and this is now reflected in the colour coding and legend (“Dimer_2”). We also added p values to panels 5D and 5E in Figure 5, and they show a clear difference in the frequency of extracellularity between trees with monomer and homomer roots. We also updated the manuscript text to reflect these changes.

Figure 4.

Figure 5.

Depending on how all this turns out, some further discussion of the ratchet view of evolution might be warranted.

Finally, a popular model for the evolution of multimers invokes a domain-swapping scenario, whereby a monomer with an internal pair of binding domains loses the hinge (by deletion) and then must swap domains with a partner to close the loop. If the data allow the identification of such multimers, it might be worth taking a closer look at the data for prokaryotes vs. eukaryotes, as Lynch has pointed out the potential barrier to dimerization by this route in diploid eukaryotes resulting from selection against heterozygotes.

In our experience, the type of full domain swapping that is described in Lynch 2012 MBE 29:1353 is rare in the PDB, although swapping a few (a dozen or two dozen) residues at the termini of the proteins is not uncommon in dimers. However, selection against heterozygotes can be bypassed in eukaryotes if homomers assemble co-translationally, which does not result in heterozygotes in “cis”, if the same mRNA is used in the process. Interestingly, a recent preprint indicates that co-translational interfaces are more “intertwined” compared to post-translational ones (<https://www.biorxiv.org/content/10.1101/2024.01.20.576408v1>), so a process that is somewhat similar to the one described by Lynch might facilitate the evolution of complex interfaces in co-translationally assembled homomers (although, the pattern does not seem to be restricted to eukaryotes). However, this pattern may also result from other mechanisms, including CNE and genetic dominance (see Badonyi and Marsh 2023 for an overview, <https://doi.org/10.1111/febs.16869>).

Reviewer #2 (Remarks to the Author):

Abrusán and Zelezniak demonstrated differences in homomer versus monomer binding interfaces, with the several conclusions on how cellular location drives the evolvability of enzyme quaternary structure. In general, the manuscript is written with great clarity and contains a conclusive and well-reasoned analysis of PDB structures showing there is an observable difference in monomers versus homomers. However, beyond the thorough analyses, I struggled to understand what the conceptual advancement was, as much of the discussion appears as rehashing literature already cited in the introduction without specific connection to the presented results. The manuscript would also benefit from less technical terms or at least a better introduction of those terms in order to be accessible for a broad readership.

We tried to make the discussion clearer and more “reader-friendly”. In our view, there are three conceptual advancements in the manuscript: **1)** Oligomerisation of enzymes is context-dependent, and is rare outside of the cell; thus, frequently, it is unlikely to be the direct consequence of the biochemical function of the protein, because that is not determined by the cellular environment. **2)** The evolution of quaternary structure indicates that neutral factors (CNE) are a major force in shaping it, and the hydrophobic ratchet is an important factor in homomer evolution. The original paper of Hochberg et al. 2020 in Nature had a brilliant insight but it did not provide evidence that CNE is actually an important process, or that interface evolution has ratchet-like characteristics. **3)** Selection pressure on quaternary structure may operate at the level of the cell and not just at the level of the protein, and it may act through shaping diffusion rates, cytoplasm fluidity, cellular osmotic pressure, compartmentalisation of metabolic processes or other factors. While we provide only indirect evidence for the last possibilities, generally, papers studying the purpose of homomerisation consider only factors that operate on the protein(complex) itself, like improved stability, catalytic efficiency, regulatory consequences and similar. We think that this is an important insight of our work, despite being only a hypothesis.

Two general points

- The introduction may actually partly be based on your results, hence why the introduction and discussion seem similar. You reference Figure 2 in the introduction on line 121, the discussion on lines 354-359 seems similar to lines 81-86. The manuscript may be more clear if you state what is currently known in the field without referencing any of your findings, and then more clearly indicating how your results specifically support or reject hypotheses on what could cause location-dependent evolution (e.g., the diffusion rate but not the Michaelis-Menten analysis from the supplement appear selected on). Lines 361-365 just state existing ideas without connecting to your results, and again blurs what is introduction and what is discussion. Lines 345 and 366 seem to justify an observation if selection favours high diffusion rate and then later state that selection remains to be seen, again making it hard to know what exactly I should take away from this paper.

We added new material to the manuscript; a new figure (Figure 7) and two new supplementary figures (Figure S10 and S11), which address the predictions of some of the hypotheses discussed in the above-mentioned sections, like the possibility that oligomerisation is the consequence of macromolecular crowding, or that it may contribute to the maintenance of cellular homeostasis. Accordingly, we changed parts of the manuscript, including the discussion, and hopefully reviewer will find it better now.

- I find the multistate versus binary tree analysis overly complicated, given the results are almost similar. Figure 4E looks interesting as a use of the multistate tree, but I find this hard to reconcile with the other results if losing an interface could be evolutionarily advantageous if lower MW/faster diffusion is favoured etc. Unless there is a more concrete point you want to make for differentiating homomer unit size, I would consider simplifying the numerous presented analyses by only using binary trees.

The binary tree analysis was used because, conceptually, it is very simple and intuitive, but we think the multistate analysis is the more correct. Additionally, having the same number of subunits does not necessarily mean having the same interfaces, and the multistate analysis can incorporate this, while the binary analysis cannot. In the revision, we changed slightly the multistate analysis and discriminated between complexes with the same number of subunits but different, non-homologous interfaces. A disadvantage of the multistate analysis is that it is more complex, and the ancestral state of the root cannot be inferred for a larger number of trees than in the case of binary analysis.

I have several more specific comments below:

- I believe the figures are cited out of order (Fig 1 is first referenced on line 259)

Figure 1 is first referenced in line 92.

- Lines 106 and 117: <https://journals.aps.org/pre/abstract/10.1103/PhysRevE.98.022113> demonstrates the likelihood of aggregation (unbound) or destabilisation (nondeterministic) behaviour in a protein quaternary structure model.

- Line 96: “whether quaternary structure influences the evolution of new functions” – I’m not sure this point was addressed, instead just compared two types.

We think this is not an overstatement in the Intro; technically, we examined whether the evolution of ligand binding (which we used as a proxy of function) is similar in different types of homomers and monomers. We initially assumed that complex formation would enable faster evolution of binding sites, due to the

faster evolution of quaternary structure than tertiary structure, however, the pattern we found was exactly the opposite of the expected.

- **Line 148, Typo in “Figures 2 ad 3” should be “and”**

We fixed this.

- **Line 156/168/170: there appears to be an inconsistent use of significant figures, unless line 156 and 157 are meant to indicate approximations**

We are not sure what reviewer means in this case.

- **Line 138, EC should be defined more explicitly for a general audience (as is almost done on line 390)**

We added the definition of the EC number.

- **Line 245 and 248: Should these refer to 4D and 4E instead?**

Reviewer is correct, we fixed the figure labelling.

- **Line 265: What is the implication here? That complexes with monomer roots are more likely to independently evolve interacting interfaces?**

Yes. In the case of trees with homomer root, the location of interfaces remains largely the same in the orthogroup. In contrast, in the case of trees with monomer roots different interfaces frequently evolve repeatedly. Since the tertiary structure within the orthogroups does not change substantially (TM-score is above 0.75-0.8 in most cases), this also means that a given fold can evolve interfaces at different places, so there might be a considerable degree of randomness in the location of actual interfaces.

- **Lines 307-322: The sample sizes here are likely too small to justify this analysis. Although you caveat the uncertainty, the results (especially in Figure 5G-L) could support the opposite interpretation if the outliers were removed (or is there any evidence hinting as to why those specific homomers evolved different sizes in the cellular locations?).**

The amount of data is indeed small, we are fully aware of this, and we state in the manuscript that the uncertainty associated with this analysis is high (lines 322-323). Unfortunately, this is not something we can change, the only alternative is to exclude these panels. The location of outliers has only a minimal effect on the statistics, and does not influence conclusions, because we use a non-parametric test (Wilcoxon). However, as the number of data points is small, their removal matters, and it matters more than their value.

- **Line 335: Could consider citing <https://doi.org/10.1371/journal.pcbi.1006886>, which also discusses the order subunits reflecting order of evolution, as well as the neutral evolution of interfaces, albeit in a toy model.**

We added this paper to the manuscript.

- **Line 365: no need for the comma before the parentheses**

Fixed.

- **Figure 4 E: typo on “decreasig”**

Fixed.

- **Figure 4 F&G: all other p-values are given exactly, whereas these give the default reporting threshold from R (2.2e-16)**

We fixed this, and now the exact values are provided. (We previously used the sasLM R package to perform Type III ANOVA, which does not seem to be able to provide exact values below 2.2e-16)

- **Figure 5 H-L: Does the p-value refer to both groups combined or are they treated separately (line 318).**

The p-value refers to both groups combined.

- **Line 381, I initially took this as signal and transit peptides were excluded from all analyses, yet they repeated appear in later methods (e.g., lines 398, 399, 447, etc). If this means you excluded signal and transit peptides from the 100 aa filtering, could you rephrase?**

We tried to improve the wording here. The presence of signal and transit peptides is important in determining the cellular location of a protein, so when we used DeepLoc2 or PSORTb to determine it, the full version of sequences containing the signal/transit peptides was used. However, everywhere else, the sequence without signal peptide was used, for example, in the sequence alignments, minimal length cutoff, or defining the coverage of the PDB files, because under normal circumstances these peptides are cleaved from the protein.

Reviewer #2 (Remarks on code availability):

The code availability statement in the manuscript is “... freely available at Zenodo (<https://zenodo.org/records/...>)”, and no code was provided to review, so I have not reviewed any, although it does not appear to be an important contribution of this paper.

At the time of submission, for review purposes, the code was submitted with the rest of the manuscript to Nature Communications, and was available for the reviewers as “Dataset 2”. Similarly, we again uploaded the revised code of the revised manuscript through the submission site of Nature Communications, so it is available for the Editors and Reviewers. The final code, if the manuscript will be accepted, will be distributed on Zenodo.

Reviewer #3 (Remarks to the Author):

This paper investigates what factors influence quaternary structure in proteins. In particular, it uses bioinformatic analyses to test if there are signals for a functional benefit of oligmerization. In general, I found this paper very interesting and quite well presented and of general interest.

I have a few comments. In general, one needs to remember that selection is hard to prove in the best cases, and the absence of it even more difficult to prove. The authors are already quite careful, but they might consider making the limitations of such large scale studies more explicit. Even very small

selection coefficients can in principle construct and maintain interactions especially in prokaryotes. I think it's important to bear this in mind.

My bigger concern are the ancestral state reconstructions. I object to two methodological choices. The trees are rooted with maximum ancestral deviation, which is a very inaccurate rooting technique, because it can be influenced by taxon sampling. Unfortunately, there is no free lunch when it comes to tree rooting - it is often the most difficult problem in prokaryotic phylogenies. At the very least, I would want this rooting method validated with more reliable methods (paralogy rooting, rooting according to the species tree, or DTL rooting) for some appreciable number of trees. I also found it hard to visually verify whether the roots of the trees in the main text are plausible because the trees not labelled sufficiently (not at all in Figure 4 and with hard to interpret taxon abbreviations in Figure 5).

We thank Reviewer for the constructive criticism. While we may differ with the Reviewer on several specifics mentioned in their comment (detailed responses follow below), we acknowledge that in essence, the Reviewer's comment relates to testing how and whether the topology of the trees (and their potential variability based on methodological choices) influences our conclusions (including the rooting, but this also means the alignment method or the tree inference algorithm). We do agree that this is a highly relevant question, and we made several checks to ensure that the conclusions are not influenced by these methodological choices (see below).

In the case of rooting, we think the choice of using the Minimum Ancestral Deviation (MAD) method was a good choice. There are a few benchmarks that compare different rooting algorithms, and these indicate that MAD generally outperforms other methods. One relatively recent one is from the Bansal lab, which generally "champions" the DTL method (rooting according to the species tree, Wade et al. 2020, PLoS One, <https://journals.plos.org/plosone/article?id=10.1371/journal.pone.0232950>). They conclude that on real, empirical datasets MAD was the best-performing method, while the DTL method performed well only on simulated trees. In addition, MAD was robust to tree reconstruction errors and high rates of horizontal transfer events, while DTL was not. Other benchmarks (Tria et al. 2017, Nature Ecol, Evol. <https://www.nature.com/articles/s41559-017-0193>, Lamarca et al. 2022, Mol. Phyl. Evol. 169:107434, <https://doi.org/10.1016/j.ympev.2022.107434>) also show that MAD generally outperforms other methods, like midpoint rooting, or outgroup rooting. (Note that MAD was developed by Tria et al. 2017, so they weren't independent).

The problems with DTL (Duplication-Transfer-Loss) rooting can be illustrated with the tree on Figure 5A (See below; the tree is somewhat unusual, in the sense that it is among the most diverged orthogroups in the dataset). DTL rooting works by comparing the gene tree to a species tree, and relies on the assumption that species trees (and their root) can usually be inferred much more accurately than trees of individual proteins.

The tree contains proteins with the FAD_binding_4 PFAM domain, and while the sequence similarity between them can be very low, the structural similarity is reasonably high, TM-score is 0.75, even for distant ones like D2HDH_HUMAN (PDB: 6lpn) and XYLO_MYCTT (PDB: 5k8e). Note that there are two different types of dimers in the tree, with non-homologous interfaces. The red clade at the bottom (between UniProt IDs: Q6NAV4_RHOPA and Q0SBK1_RHOJR) contains microbial proteins and proteins from eukaryotic organelles (mitochondrion and peroxisome, the peroxisome has several proteins of mitochondrial origin, and even its endosymbiont origin cannot be completely ruled out). These eukaryotic proteins clearly group with microbial ones both in terms of structure and sequence, but the DTL method would struggle with this, due to the lack of clear separation between eukaryotic and prokaryotic taxa. Also, there are other prokaryotes in the phylogeny (clades D3Y1I2_9ACTN – HEXNO_RALSU; DPRE1_MYCTU – XYOA_STRCO), thus trying to use the Eukaryotic proteins only and root them with the prokaryotic ones would be far from trivial.

In such situations (prokaryotic and eukaryotic proteins are present in the same tree; see also Figure 1 of the manuscript) it is also unclear what outgroup one should use, in case one would decide to use the outgroup method. Of the 311 trees we have, 243 trees have proteins from more than one kingdom, 92 from all three kingdoms, and 78 contain proteins from an organelle that has (at least some) proteins of endosymbiont origin (Mitochondria, Chloroplast, Peroxisome). Like in the tree above, they do not always form distinct clades, indicating that these ancient enzymes underwent lots of horizontal transfer events.

We think MAD was a good choice not just because it is quite accurate, but because it can be applied to all trees, and one does not have to make assumptions about the “right” outgroup, or about the topology of the species tree. These things can be difficult or even impossible to resolve, and the orthogroups would be affected differently by the ad hoc choices one would have to make.

In addition to MAD, we examined whether a different commonly used rooting method, the so-called “midpoint rooting” influences the results. Similarly to MAD, midpoint rooting can be applied to all trees, and is relatively resistant to errors in the tree reconstruction, but it is more sensitive than MAD for heterogeneous evolutionary rates in the tree. We found that using it does not change the conclusions, see the version of Figure 4 below, which was made using trees rooted with the midpoint method (note that we changed the order of the panels):

An indirect way of assessing the accuracy of the root is, if one uses a different alignment method to make the trees. This can change not just the root, but also the broader topology of the tree. In the manuscript we used Mafft-DASH because it utilises structural information in the alignments, and for this dataset where all sequences have a structure it is probably one of the most accurate methods available. Recently an update to the popular muscle tool was published (Muscle5) which has good accuracy also on distantly related sequences (Edgar, 2022, Nature Comms., <https://doi.org/10.1038/s41467-022-34630-w>). Since it is using only sequence information it is nevertheless inevitably less accurate than Mafft-DASH. We reanalysed the data using Muscle5 alignments AND midpoint rooting, and it did not result in a change in the pattern or conclusions (see below a version of Figure 4 made with Muscle5 and midpoint rooting):

Similarly, the conclusions based on Figure 5 also do not change when Muscle5 alignments and midpoint rooting are used:

Overall, we think that the results of the phylogenetic analysis are robust to variations in the topology of the trees, be it due to the method one uses to root the trees, or construct alignments (or to infer the trees themselves, which we did not show here.)

However, the tree in Figure 5A is affected by the choice of aligner and rooting method, which is summarized below. From the four possibilities, three indicate a monomer root, and one a homomer root.

In retrospect, this particular tree was probably not a perfect choice for Figure 5A, because its root status depends on the method used. However, its only purpose is illustration; originally it was chosen for aesthetic reasons, and because it demonstrates several aspects of the results well: the clustering of similar quaternary structures, and that extracellular proteins “reside” in the monomer part of the tree. For this reason, we decided to keep it in the manuscript.

Finally, for Figure 5 one can also use an alternative method, that to some degree bypasses the rooting problem - one can use the ancestral state of the smallest subtree that contains all extracellular proteins. (In the case of the trees above, this would be the subtree encompassing all monomers, except DLD_ECOLI). We checked this as well, and it resulted in similar patterns and conclusions as using the root (not shown). However, this is only a partial solution, because for many trees this smallest subtree is the same as the full tree.

This brings me to the biggest problem: The paper estimates ancestral states at the roots of their phylogenetic trees using a likelihood method. I'm afraid this is indefensible. The root position along the root branch is not known, but usually arbitrarily assigned by tree analysis software (often in the middle of the branch). A maximum likelihood estimate uses this position, even though there is no data to determine it. This is a very unfortunate and frequent issue in papers that use this method and it simply should not be done. The problem disappears when using parsimony, because parsimony does not use the branch lengths to estimate ancestral states. Unfortunately I cannot sign off on maximum likelihood root reconstructions.

We reanalysed the results of Figure 4 and Figure 5, with maximum parsimony (MP) ancestral state estimates (using the Phangorn R package, and the accelerated transformation algorithm), both in the case of phylogenies made with Mafft-DASH and Muscle5. The results are very similar to the results we got with maximum likelihood (ML), and none of the conclusions are affected (see the figures below).

Recent benchmarks comparing the ML and MP methods (<https://www.biorxiv.org/content/10.1101/2023.08.31.555762v2>) indicate, that there is no “best” method for ancestral character estimation; MP does not perform well on trees with long branches, while ML can be overparametrized when the All Rates Different (ARD) model is used. However, according to the author of the benchmark, the Equal Rates (ER) ML model (which we used in the manuscript) is robust, and performs well even when the trees are small, or the assumption of equal rates is violated in the character evolution, and it is probably the preferred general-purpose method.

Figure 4:
Mafft-DASH, MAD rooting, maximum parsimony:

Muscle5, MAD rooting, maximum parsimony:

Similarly, the conclusions of Figure 5 as also not affected by using maximum parsimony, irrespectively whether Mafft-DASH or Muscle5 alignments were used.

Mafft-DASH, MAD, maximum parsimony:

Muscle5, MAD, maximum parsimony:

The low significance on panels B above is largely the consequence that compared to ML, even fewer trees having extracellular proteins have homomer roots (only 4 and 5; see panels D above). Importantly, despite the small number of such trees, the difference between the two root types on panels D and E remains highly significant.

In the case of the tree on Figure 5A, the ancestral state of the root does depend on the ancestral state reconstruction and alignment method:

Mafft-DASH, MAD, MP

Muscle5, MAD, MP

Muscle5, Midpoint, MP

For me to recommend publication of this very interesting and valuable contribution the issues with rooting and ancestral state estimation will have to be resolved.

Overall, we think that our results are robust, and our conclusions are not affected qualitatively by the methodological choices involving the phylogenies, or ancestral state reconstruction. Given the benchmarking information available in the literature and our own tests, we believe our choices of the methods and tools were reasonable, and most likely better than the alternatives.

REVIEWER COMMENTS

Reviewer #1 (Remarks to the Author):

I am reasonably satisfied with the authors' responses, although do not entirely agree with what they say.

In response to my query about equilibrium, the response seems a bit confused. Maybe they did not understand my point (which I didn't think was that complicated). They continue to seem to believe that a ratchet is occurring, without actually testing for this.

In their first paragraph in response (where they refer to Hochberg), for example, they argue that gains can be due to two forces but losses only due to selection, and also invoke mutation bias. However, none of these issues have anything to do with the existence of an equilibrium level of divergence – the relative strengths of forces influence the position of equilibrium distribution, but there must always be an equilibrium resulting from opposing forces, even if everything were driven by just drift and mutation.

Likewise, in the paragraph starting with "Second, equilibrium...." the authors state that the equilibrium depends on the numbers of trees. This also doesn't make any sense to me. The number of trees examined might influence the authors' ability to test for equilibrium, but has nothing to do with its presence or absence. The simplest test for equilibrium would be to evaluate whether along the more recent branches in phylogenies there are equal total numbers of gains and losses. The most common state would be so because it experiences a lot of gain and/or little loss, whereas the rarest states would experience a lot of loss and/or little gain (keeping them rare). This gets a bit complicated, as the authors now note that different species with the same complexity of multimeric state can have different interfaces, which would mean both losses and gains even though at face value it would appear to be a static system.

Perhaps the authors could clean this up a bit.

Reviewer #2 (Remarks to the Author):

My major concerns have been addressed during revision, and the manuscript has been strengthened after adding analyses requested by the other reviewers.

Reviewer #2 (Remarks on code availability):

The code provided is sufficient to recreate the figures provided in the manuscript.

Reviewer #3 (Remarks to the Author):

The authors have now shown parsimony results in their response to reviewers, but not the actual manuscript. They have also only addressed concerns about the accuracy in the roots in the response as far as I can tell.

I will re-iterate my concern here once more: Root reconstructions are wrong if using a likelihood method. They really are unacceptable, as one of the parameters needed for the calculation (the exact position of the root node along the root branch) is not known and usually just arbitrarily assigned. This is wrong and we need to stop tolerating this in manuscripts. Parsimony is the only way the authors can make their claims about root reconstructions. Citing papers about the accuracy of likelihood methods is no way out of this dilemma, because those papers are concerned with 'normal' nodes of the trees, not the root node. I can only re-iterate that this is a fairly elementary mistake that I will not endorse for publication. It's simply wrong. More importantly - the fix is obvious: Just present the data using parsimony. As was made clear in the response: The conclusions remain the same. Please: Just do this right and set a good example instead of propagating a methodological fallacy that already pollutes our field. I see no disadvantage in just showing the parsimony results. Just to be clear: I would have no problem with the likelihood methods for anything OTHER than the root node.

To be clear: I am otherwise quite enthusiastic about this paper - the authors did great work. It just isn't worth digging about a methodological choice (root reconstructions using a likelihood model) that makes no sense and is obviously wrong to anyone familiar with ancestral state reconstruction.

About the rooting methods: I wasn't terrible satisfied with the response. What I am looking for is at least one or two examples of trees that the authors root manually and depict in a way the reader can actually follow. That means actually checking if their automatically calculated roots are biologically plausible in terms of HGT and duplications.

A few more small things: The authors refer to bacteria eukaryotes and Archaea as kingdoms. This is very outdated terminology and they should call them domains. In addition, they refer to proteins as 'older' if they find them in all three domains. This is only true if inheritance is vertical. To make that claim, they would have to check whether inheritance really is mostly vertical and whether these proteins existed in LUCA or the LCA of Bacteria, say. A little bit of manual effort goes a long way in making such claims more trustworthy.

I have no interest in a protracted fight about this paper, as I find it generally interesting. I would plea with the authors either to replace the likelihood root reconstructions with parsimony reconstructions in the paper or to at least write a prominently placed paragraph that explains that what they did is formally wrong, is only done for expedience, and makes arbitrary assumptions

about. a key parameter of the reconstructions.

REVIEWER COMMENTS

Reviewer #1 (Remarks to the Author):

I am reasonably satisfied with the authors' responses, although do not entirely agree with what they say.

In response to my query about equilibrium, the response seems a bit confused. Maybe they did not understand my point (which I didn't think was that complicated). They continue to seem to believe that a ratchet is occurring, without actually testing for this.

In their first paragraph in response (where they refer to Hochberg), for example, they argue that gains can be due to two forces but losses only due to selection, and also invoke mutation bias. However, none of these issues have anything to do with the existence of an equilibrium level of divergence – the relative strengths of forces influence the position of equilibrium distribution, but there must always be an equilibrium resulting from opposing forces, even if everything were driven by just drift and mutation.

Likewise, in the paragraph starting with "Second, equilibrium..." the authors state that the equilibrium depends on the numbers of trees. This also doesn't make any sense to me. The number of trees examined might influence the authors' ability to test for equilibrium, but has nothing to do with its presence or absence. The simplest test for equilibrium would be to evaluate whether along the more recent branches in phylogenies there are equal total numbers of gains and losses. The most common state would be so because it experiences a lot of gain and/or little loss, whereas the rarest states would experience a lot of loss and/or little gain (keeping them rare). This gets a bit complicated, as the authors now note that different species with the same complexity of multimeric state can have different interfaces, which would mean both losses and gains even though at face value it would appear to be a static system.

Perhaps the authors could clean this up a bit.

Indeed, we do believe a ratchet is occurring, and among other things, we do show that interfaces are gained easier than lost, which indicates the existence of a ratchet. In the revised version of the manuscript, we also discuss the possibility of an additional mechanism that may act as a ratchet: protein degradation, which, together with macromolecular crowding, can result in positive feedback between the two. Besides the hydrophobic ratchet, it may be an additional factor driving the evolution of interfaces (although we are not testing this explicitly in this manuscript).

However, showing that there is a clear bias between interface gains and losses is not the same as testing whether the "protein world" is in an equilibrium from the perspective of interfaces, as reviewer suggested. That was not our goal, and we think that this is indeed an interesting hypothesis, deserving a study on its own. This, however, would be challenging potentially requiring quantifying of several processes, including the rates of

evolution of heteromers from homomers/monomers, the number of trees, etc. Instead, in our work, we hypothesise that protein complex formation does contribute to cellular homeostasis, and that suggests that the homomer/monomer ratios are optimized to some degree in the cells, and are likely to be stable (which, as Figure S11 shows is pretty much the case). However, this does not automatically mean being in an equilibrium when the entire “protein world” is considered, although it might mean a state close to equilibrium (at least from the perspective of the number of interfaces/particles) in a particular genome.

In addition, when the entire history of life is examined, it is characterised by a gradually increasing complexity of organisms over time, without clear signs of reaching a plateau (~equilibrium), and we see no reasons to assume that this was not the case also from the perspective of their biochemistry, i.e. more complex pathways, molecular machines, etc. appearing over time.

Reviewer #2 (Remarks to the Author):

My major concerns have been addressed during revision, and the manuscript has been strengthened after adding analyses requested by the other reviewers.

Reviewer #2 (Remarks on code availability):

The code provided is sufficient to recreate the figures provided in the manuscript.

Reviewer #3 (Remarks to the Author):

The authors have now shown parsimony results in their response to reviewers, but not the actual manuscript. They have also only addressed concerns about the accuracy in the roots in the response as far as I can tell.

I will re-iterate my concern here once more: Root reconstructions are wrong if using a likelihood method. They really are unacceptable, as one of the parameters needed for the calculation (the exact position of the root node along the root branch) is not known and usually just arbitrarily assigned. This is wrong and we need to stop tolerating this in manuscripts. Parsimony is the only way the authors can make their claims about root reconstructions. Citing papers about the accuracy of likelihood methods is no way out of this dilemma, because those papers are concerned with 'normal' nodes of the trees, not the root node. I can only re-iterate that this is a fairly elementary mistake that I will not endorse for publication. It's simply wrong. More importantly - the fix is obvious: Just present the data using parsimony. As was made clear in the response: The conclusions remain the same. Please: Just do this right and set a good example instead of propagating a methodological fallacy that already pollutes our field. I see no disadvantage in just showing the parsimony results. Just to be clear: I would have no problem with the likelihood methods for anything OTHER than the root node.

To be clear: I am otherwise quite enthusiastic about this paper - the authors did great work. It just isn't worth digging about a methodological choice (root reconstructions using a likelihood model) that makes no sense and is obviously wrong to anyone familiar with ancestral state reconstruction.

We appreciate the reviewer's constructive comments and criticism, but we do not entirely share his/her view here. We are aware that estimating the ancestral state of the root node is more difficult than for other nodes, however parsimony methods also have several shortcomings, perhaps the most important one from our perspective is that they do not utilise the branch lengths of the tree the same way as likelihood methods do. Since the estimation of the state of root depends also on the accuracy of the estimation of ancestral states at other nodes, this can have a significant effect on the root estimates. A brief description of the shortcomings of maximum parsimony for ancestral state reconstructions is available at

<https://journals.plos.org/ploscompbiol/article?id=10.1371/journal.pcbi.1004763>.

To check which method to use, we performed a small simulation study, using the trees of the manuscript. We simulated the evolution of a binary discrete variable on our empirical trees with the geiger R package, with different transition rates (higher rates result in faster evolution of characters along the tree). The sim.char function was used, with a discrete model, the Q-matrix was defined as $q \leftarrow \text{list}(\text{rbind}(c(-\text{rate}, \text{rate}), c(\text{rate}, -\text{rate})))$. For every tree ten replicates were made, to account for variability between the simulations.

Thus, the ancestral state of the root was known for every tree, and we examined the accuracy of its reconstruction using the simulated character states at the tree leaves, both with maximum parsimony (MP) and maximum likelihood (ML). The results show that ML consistently outperforms MP, irrespectively of the transition rates used:

The transition rates were set in two ways: **A)** we used the same, fixed rate for all trees, and **B)** we also used scaled rates, where the rate for every tree was defined individually, as the average distance of the leaves from the root, divided by a scaling factor (50 to 5). The same ancestral reconstruction methods were used as in the manuscript, i.e. the ace

function of the APE R package for ML (ER model), and the `ancestral.pars` function of the `phangorn` R package for MP (ACCTRAN model).

To some degree, these simulations can also be used to make an estimate of the accuracy of the reconstruction in the binary analysis: in the trees where the ancestral state of the root could be estimated (i.e. its probability was higher than 0.51) the median|mean transition rate obtained with the APE package is 0.161|0.236, and the median|mean scaling factor is 10.9|18.4, so the accuracy of the root state is expected to be ~90% with ML. However, our trees do contain horizontal transfers, and the transition rates estimated with the APE package tend to be somewhat higher than the rates that were used in the simulations with the Geiger package, so this is a coarse estimate.

It is beyond the scope of this paper to find out which differences between the implementations of the ML and MP algorithms we used are responsible for the consistently better performance of ML. We also did not simulate complex evolutionary scenarios involving horizontal transfers, or tested how the pattern depends on tree topology, tree size, or other factors. We are also aware that the `geiger` package simulates character evolution with a Markov process (most packages do, this seems to be the standard way of simulating the evolution of discrete characters along a tree). Nevertheless, these results are in agreement with the general perception in the field that ML is more accurate than MP, and therefore we decided to keep the results based on ML reconstruction in the main text.

However, we also added the results obtained with MP to the supplementary materials, so that the results are available with both methods, and readers who prefer (or trust) MP more have access to them.

About the rooting methods: I wasn't terrible satisfied with the response. What I am looking for is at least one or two examples of trees that the authors root manually and depict in a way the reader can actually follow. That means actually checking if their automatically calculated roots are biologically plausible in terms of HGT and duplications.

We think benchmarking the MAD rooting method is beyond the scope of the manuscript, as it is an established method that has been benchmarked several times in the past. It is known to be one of the best performing rooting methods (see the benchmarks in: <https://journals.plos.org/plosone/article?id=10.1371/journal.pone.0232950>, <https://www.nature.com/articles/s41559-017-0193>, <https://doi.org/10.1016/j.ympcv.2022.107434>).

Also, selecting one or two examples to demonstrate that the method “works” would be dangerous practice subject to research bias. This is why we previously reanalysed the entire dataset (see our previous response), and showed that changing the alignment method, rooting method, or ancestral reconstruction method does not affect the results, indicating that our results are robust to variations in tree topology and methodological choices.

As every rooting method (including the outgroup method) MAD is not infallible, and we show below one case where we think it is correct, and one case where it is likely to be erroneous.

An example where MAD is correct is Dihydrofolate reductase. Bacterial and eukaryotic proteins from the PDB form separate clades (see below):

The archaeal protein DYRA_HALVD is likely to be the result of an ancient horizontal transfer, (BLASTing it against the entire NR indicates that its closest bacterial homologs are more similar to it than its eukaryotic homologs). Re-building the tree only for prokaryotes, and rooting it with MAD results in essentially the same root as with the root with the eukaryotic outgroup, indicating that MAD works well:

A different example where MAD is unlikely to be correct is the phylogeny of Ketol-acid reductoisomerase. The tree we used also has a prokaryotic and eukaryotic branch:

When the eukaryotes are removed, and the tree is re-built and re-rooted with MAD for prokaryotes, it results in a comparable topology, however the root of the prokaryotic branch is placed at a different node than when the eukaryotes are present:

This is partly the consequence of the low bootstrap support of some of the internal nodes (note that the tree below is unrooted):

However, this tree has also another problem: the few eukaryotic proteins of the tree are from chloroplasts (ILV5_SPIOL, ILV5_ORYSJ) and mitochondria (A0A2H0ZMH9_CANAR), and they are of bacterial (endosymbiont) and not archaeal origin. Thus, their closest homologs are almost certainly bacterial, and the position of the eukaryotic clade is likely to be incorrect. (The fact that proteins of mitochondrial and chloroplast origins are in the same clade indicate that it is almost as old as the last bacterial common ancestor.) Ketol-acid reductoisomerases have two forms, a short (class I) and a longer (class II) form, which among other things differ in C terminal duplications, but also in non-duplicated sequence. The placement of the eukaryotic clade improves if the alignments are pre-processed, and the additional sequences that characterise the (class II) ILVC_ECOLI and the (class II) eukaryotic proteins are removed. This results in placing the eukaryotic proteins closest to the E. coli branch, and a biologically more meaningful topology that places the class II proteins closer (see below, note that the tree is unrooted):

The unambiguous identification of the position of the root might be very challenging for these proteins (especially for an automated method), as they have a complex evolutionary history (see also doi:10.3389/fpls.2017.01540), and only a limited number of structures in the PDB. However, for this particular orthogroup, the differences in tree topology

appear to have no effect on the results, as the ancestral state of essentially all internal nodes are dimers with the same, homologous interface.

Previously we did not perform any alignment preprocessing, and in some cases, it might result in more accurate topology of the trees. Motivated by this case, we decided to repeat the phylogenetic analyses also with preprocessed alignments, where the alignments were trimmed, and columns with 75% or more gaps were removed. The results show that this does not change the patterns, or any of our conclusions (see below).

Figure 4, with trimmed alignments (MAD rooting, ML ancestral character reconstruction):

Figure 5, with trimmed alignments (MAD rooting, ML ancestral character reconstruction):

We decided to keep the results based on the unaltered alignments in the manuscript.

A few more small things: The authors refer to bacteria eukaryotes and Archaea as kingdoms. This is very outdated terminology and they should call them domains. In addition, they refer to proteins as 'older' if they find them in all three domains. This is only true if inheritance is vertical. To make that claim, they would have to check whether inheritance really is mostly vertical and whether these proteins existed in LUCA or the LCA of Bacteria, say. A little bit of manual effort goes a long way in making such claims more trustworthy.

We changed the terminology as suggested (to domains) wherever possible, however prokaryotes and eukaryotes are morphological categories, and not taxonomic groups, so we also use the simplest “group” term to avoid using the really outdated “empire” terminology.

I have no interest in a protracted fight about this paper, as I find it generally interesting. I would plea with the authors either to replace the likelihood root reconstructions with parsimony reconstructions in the paper or to at least write a

prominently placed paragraph that explains that what they did is formally wrong, is only done for expedience, and makes arbitrary assumptions about a key parameter of the reconstructions.

We included the maximum parsimony results to the supplementary materials so that both MP and ML results are available for the readers. However, we didn't include the above simulation results (which were admittedly simple) in the manuscript; Nature Communications also publishes the responses for the reviewers; thus, this information is available to the readers.

REVIEWER COMMENTS

Reviewer #3 (Remarks to the Author):

My main concern is not sufficiently addressed. Their simulations still assume a known root node position along the root branch, in which case ML is of course more accurate than parsimony. The problem is that this key parameter (position of the root node along root branch) is not known in a real tree, but arbitrarily assigned. Their concerns regarding parsimony are not supported by the reference provided, which suggests branch lengths and rate variation are the primary concerns. Parsimony is not equivalent to a joint ancestral state reconstruction.

The inclusion of parsimony in the paper satisfies me, even if the response does not.